# Six-Derivative Yang-Mills Couplings
# in Heterotic String Theory

Mohammad R. Garousi[1]

*Department of Physics, Faculty of Science, Ferdowsi University of Mashhad*
*P.O. Box 1436, Mashhad, Iran*

## Abstract

In this work, we present a comprehensive analysis of the structure of six-derivative bosonic couplings in heterotic string theory. First, we determine the maximal covariant and Yang-Mills gauge invariant basis, which consists of 801 independent coupling constants. By imposing T-duality constraints on the circular reduction of these terms, we obtain 468 relations between the coupling constants at the six-derivative order and the known couplings at lower derivative orders. Through the use of field redefinitions, we are able to eliminate the remaining 333 coupling constants. Remarkably, we find that the Yang-Mills field strength only appears through the trace of two field strengths or their derivatives. Finally, we perform further field redefinition to rewrite the remaining couplings in a canonical form characterized by 85 independent couplings.

---

[1]garousi@um.ac.ir

# 1   Introduction

Classical string theory is known to exhibit T-duality to all orders in derivatives, as demonstrated in previous studies [1, 2, 3]. This powerful T-duality symmetry has been recently leveraged to establish that all covariant and Yang-Mills (YM) gauge invariant couplings involving an odd number of derivatives in heterotic string theory vanish [4]. Furthermore, this symmetry has been used in [5] to include the YM couplings at the four-derivative order in the covariant Metsaev-Tseytlin and Meissner actions. The four-derivative couplings in heterotic string theory have also been determined within the frame-like Double Field Theory approach [6, 7], which reproduces the couplings found in the non-covariant Bergshoeff-de Roo action. Building upon these previous insights, in this work we utilize the T-duality symmetry to determine the complete set of six-derivative order covariant couplings in heterotic string theory.

To construct higher-derivative couplings by T-duality, one should first find the appropriate basis for the covariant couplings with unknown coupling constants, and then fix the coupling constants by imposing the non-geometric $O(1, 1, \mathbb{Z})$ symmetry on their circular reduction [8]. The basis may be a minimal basis, in which redundancy due to field redefinitions, integration by parts, and various Bianchi identities are removed [9], or the maximal basis, in which the redundancy due only to integration by parts and various Bianchi identities are removed. In the absence of YM fields, one can impose T-duality on either basis. If one imposes it on the minimal basis, then T-duality fixes all coupling constants, and the results are consistent with S-matrix elements [8, 10, 11, 12]. If one imposes it on the maximal basis, then T-duality produces the same number of constraints between the coupling constants as in the minimal basis [8, 13][2]. However, in this case, there remain some unfixed parameters indicating there are some extra T-duality invariant couplings in addition to the T-duality invariant couplings in the minimal basis. These extra T-duality invariant couplings can be removed by appropriate field redefinitions. In fact, any set of Neveu-Schwarz-Neveu-Schwarz (NS-NS) couplings which are removable by field redefinitions are invariant under T-duality.

In the presence of YM fields, however, the number of constraints that T-duality imposes on the maximal basis at 5-derivative order and higher is greater than the number of couplings in the minimal basis. This reflects the fact that any set of couplings involving the YM and NS-NS fields which can be removed by field redefinition may not be invariant under T-duality. Previous work has identified examples of such couplings at the 5-derivative order [4]. We expect there to be similar couplings at all higher orders as well.

In other words, the NS-NS couplings in any scheme are invariant under T-duality with appropriate higher-derivative corrections to the Buscher rules [8]. In contrast, the combination of the NS-NS and YM couplings in any arbitrary scheme may not be invariant under T-duality. Therefore, in the presence of YM fields in more than four-derivative couplings, one should impose T-duality on the maximal basis to establish relations between the coupling constants, and

---

[2]Note that the number of relations between the couplings in the maximal basis in [8] is 8, which is one relation more than those in the minimal basis. This is a consequence of the fact that the most general correction to the Buscher rule was not considered for $\Delta \bar{H}$. If one considers all possible corrections to the Buscher rules, then one would find 7 relations between the couplings, as in the minimal basis.

then remove the remaining parameters by field redefinitions [4]. After removing the remaining parameters, the action may not be in a suitable form. However, it will be invariant under T-duality with appropriate corrections to the Buscher rules. One may then use further field redefinitions on the resulting couplings to rewrite them in various other schemes. The final result may not be invariant under T-duality with any specific correction to the Buscher rules. However, it will be physically equivalent to the T-duality invariant couplings and should be consistent with the S-matrix elements.

At the four-derivative order and in the presence of YM fields, the number of constraints that T-duality produces on the maximal basis is the same as the number of constraints in the minimal basis. In both cases, there are 24 constraints [5]. However, in that case, if one imposes the T-duality constraints on the maximal basis, which has 42 couplings, and on the $H\Omega$ coupling resulting from the Green-Schwarz mechanism [14], one finds there are 18 unfixed parameters. These parameters are not all removable by field redefinitions. One should impose an additional constraint that the remaining parameters must be removable by field redefinitions. This produces one extra relation between the coupling constants. Hence, even at the 4-derivative order, there are 24 relations in the minimal basis, whereas there are 25 relations in the maximal basis. Imposing these 25 relations on the maximal basis, one finds the effective action which has 17 arbitrary parameters that are removable by field redefinitions [5]. In this paper, we are going to extend the above calculations to the six-derivative order.

The paper is structured as follows: In Section 2, we find the maximal basis that incorporates the NS-NS and YM field strengths at 6-derivative order, and remove the redundancy due to integration by parts and Bianchi identities. This basis has 801 terms with unfixed coupling constants. In Section 3, we employ the T-duality technique to explicitly determine the coupling constants in the basis. We find that T-duality produces 468 relations between these coupling constants, and the fixed numbers resulting from the 2-derivative corrections to the Buscher rules that have been found in [5], and from fixed couplings resulting from the Green-Schwarz mechanism [14] that replaces $H \rightarrow H - (3\alpha'/2)\Omega$ into the 2- and 4-derivative couplings. We choose the 4-derivative coupling to be the Meissner action in which the YM fields are included [5]. We find that the remaining 333 parameters can be removed by using field redefinitions. The result is 260 couplings with fixed coupling constants that are invariant under T-duality. The YM field strength $F$ appears in the couplings only as the trace of two $F$'s or their derivatives. These couplings are in a non-standard form, which includes derivatives of the dilaton, derivatives of the Riemann curvature, and the second derivative of the $H$-field and $F$-field. They also have Ricci and scalar curvatures, as well as two-field and three-field couplings. In Section 4, we use a basis with 468 couplings which have no Ricci and scalar curvatures, no couplings with derivatives of the Riemann curvature, and no second derivatives of the $H$-field and $F$-field. We impose T-duality on this basis to fix its coupling constants. In this case, we found 107 non-zero couplings. We observe that they are the same as the 260 couplings, up to some field redefinition. These couplings are also not in a standard form because they have three-field couplings. In Section 5, we use field redefinitions on 260 or 107 couplings to rewrite them in a canonical form, in which the dilaton appears only as the overall factor $e^{-2\Phi}$, and the derivatives of the Riemann curvature, the second derivative of the $H$-field and $F$-field, and the Ricci tensor and

Ricci scalar as well as three-field couplings are removed. We could write the couplings in terms of 85 couplings. The couplings with the structure $\mathrm{Tr}(FF)R^2$ are the same as the couplings resulting from $(\mathrm{Tr}(FF) - R^2)^2$ that were found by the S-matrix method long ago [15, 16]. Section 6 provides a concise discussion of our findings and their implications. Throughout our calculations, we utilize the "xAct" package [17] for computational purposes.

## 2 Maximal basis

To construct the maximal basis, one should first consider all contractions of the NS-NS and YM field strengths and their derivatives at six-derivative order. This results in a total of 2980 such couplings. However, there is redundancy in these couplings due to integration by parts and the use of Bianchi identities. To remove the redundancy due to integration by parts, following [18], one should include all 6-derivative total derivative terms constructed from the YM and NS-NS field strengths with arbitrary coefficients to the 2980 couplings.

To remove the redundancy due to the Bianchi identities, we use the covariance and gauge invariance of the couplings to employ local frames: In the external space, a local frame can be used in which the Levi-Civita connection is zero, but its derivatives are not [18]. In the internal space, a local frame can be used in which the YM connection is zero, but its derivatives are not [4].

In the internal space local frame, the derivatives on the YM field strength become ordinary covariant derivatives [4], and the YM field strength becomes:

$$F_{\mu\nu}{}^{ij} \;=\; \partial_\mu A_\nu{}^{ij} - \partial_\nu A_\mu{}^{ij}. \tag{1}$$

Here, the YM gauge field is defined as $A_\mu{}^{ij} = A_\mu{}^I (\lambda^I)^{ij}$, where the antisymmetric matrices $(\lambda^I)^{ij}$ represent the adjoint representation of the gauge group $SO(32)$ or $E_8 \times E_8$ with the normalization $(\lambda^I)_{ij}(\lambda^J)^{ij} = \delta^{IJ}$. It satisfies the following Bianchi identity:

$$\partial_{[\alpha} F_{\beta\mu]}{}^{ij} \;=\; 0 \,. \tag{2}$$

In order to impose the above Bianchi identity into the 2980 couplings, one can write those couplings that have a derivative of $F$, in terms of the YM field $A$.

The $H$-field strength without its Lorentz Chern-Simons contribution is [19, 15]:

$$H_{\mu\nu\rho} \;=\; 3\partial_{[\mu} B_{\nu\rho]} - \frac{3}{2} A_{[\mu}{}^{ij} F_{\nu\rho]ij}, \tag{3}$$

which satisfies the following Bianchi identity:

$$\partial_{[\alpha} H_{\beta\mu\nu]} + \frac{3}{4} F_{[\alpha\beta}{}^{ij} F_{\mu\nu]ij} \;=\; 0 \,. \tag{4}$$

To impose this Bianchi identity, one can define the terms on the left-hand side of the above equation as a 4-form, and then make all contractions of this 4-form and its derivatives with $H$,

$F$, $R$, $\nabla\Phi$ and their derivatives to make six-derivative couplings. They are then added with arbitrary coefficients to the 2980 couplings.

To impose the Bianchi identities corresponding to the Riemann curvatures and the covariant derivatives, one writes all the couplings in the external local frame and writes the derivatives of the Levi-Civita connection in terms of derivatives of the metric. Then, following the same steps as in [18], one finds there are 801 independent couplings. These couplings, in a particular scheme which has no second derivative of Riemann, Ricci or scalar curvatures, no third derivative of $H$ and $F$, and no fourth derivative of the dilaton, is:

$$
\begin{aligned}
\mathbf{S_1}^{(2)} = {}& -\frac{2\alpha'^2}{\kappa^2} \int d^{10}x \sqrt{-G}\, e^{-2\Phi} \Big[ c_1 F_\alpha{}^{\gamma kl} F^{\alpha\beta ij} F_\beta{}^{\delta mn} F_\gamma{}^\epsilon{}_{mn} F_\delta{}^\varepsilon{}_{kl} F_{\epsilon\varepsilon ij} \\
& + c_2 F_\alpha{}^{\gamma kl} F^{\alpha\beta ij} F_\beta{}^\delta{}_k{}^m F_\gamma{}^\epsilon{}_m{}^n F_\delta{}^\varepsilon{}_{ln} F_{\epsilon\varepsilon ij} + c_3 F_\alpha{}^{\gamma kl} F^{\alpha\beta ij} F_\beta{}^\delta{}_{kl} F_\gamma{}^{\epsilon mn} F_\delta{}^\varepsilon{}_{mn} F_{\epsilon\varepsilon ij} + \cdots \\
& + c_{798} H_{\alpha\beta}{}^\delta H^{\alpha\beta\gamma} \nabla_\varepsilon H_{\delta\epsilon\mu} \nabla^\mu H_\gamma{}^{\epsilon\varepsilon} + c_{799} H_{\alpha\beta}{}^\delta H^{\alpha\beta\gamma} \nabla_\mu H_{\delta\epsilon\varepsilon} \nabla^\mu H_\gamma{}^{\epsilon\varepsilon} \\
& + c_{800} H_{\alpha\beta\gamma} H^{\alpha\beta\gamma} \nabla_\varepsilon H_{\delta\epsilon\mu} \nabla^\mu H^{\delta\epsilon\varepsilon} + c_{801} H_{\alpha\beta\gamma} H^{\alpha\beta\gamma} \nabla_\mu H_{\delta\epsilon\varepsilon} \nabla^\mu H^{\delta\epsilon\varepsilon} \Big].
\end{aligned}
\tag{5}
$$

The expression above represents a subset of the 801 independent couplings, with the ellipsis symbolizing an additional 794 terms that are not explicitly listed.

If we had removed the redundancy of field redefinitions as well, which would require including the following terms to the original 2980 terms [5]:

$$
\begin{aligned}
\mathcal{K}_1 \equiv {}& (\frac{1}{2}\nabla_\gamma H^{\alpha\beta\gamma} - H^{\alpha\beta}{}_\gamma \nabla^\gamma \Phi)\delta B_{\alpha\beta} \\
& - (\nabla^\beta F_{\alpha\beta}{}^{ij} - 2F_{\alpha\beta}{}^{ij}\nabla^\beta \Phi - \frac{1}{2}F^{\beta\mu ij} H_{\alpha\beta\mu})\delta A^\alpha{}_{ij} \\
& - (R^{\alpha\beta} - \frac{1}{4}H^{\alpha\gamma\delta} H^\beta{}_{\gamma\delta} + 2\nabla^\beta\nabla^\alpha\Phi - \frac{1}{2}F^{\alpha\mu ij} F^\beta{}_{\mu ij})\delta G_{\alpha\beta} \\
& - 2(R - \frac{1}{12}H_{\alpha\beta\gamma}H^{\alpha\beta\gamma} + 4\nabla_\alpha\nabla^\alpha\Phi - 4\nabla_\alpha\Phi\nabla^\alpha\Phi - \frac{1}{4}F_{\alpha\beta ij}F^{\alpha\beta ij})(\delta\Phi - \frac{1}{4}\delta G^\mu{}_\mu),
\end{aligned}
\tag{6}
$$

where the perturbations $\delta G_{\mu\nu}, \delta B_{\mu\nu}, \delta\Phi, \delta A_a{}^{ij}$ are constructed from the NS-NS and YM fields at the four-derivative order with arbitrary coefficients, then one would find the independent couplings in the minimal basis, which has 435 couplings. However, we are not interested in the minimal basis because, as we will see, T-duality produces more than 435 relations between the coupling constants in the maximal basis[3].

One may use field redefinitions to study which couplings in (5) are unambiguous, that is, they are invariant under the field redefinitions. We find there are 83 couplings in (5) that are unambiguous, and all others are ambiguous and are changed under the field redefinitions. The

---

[3]In fact, we first found the couplings in the minimal basis and observed that they are not fully consistent with T-duality. This indicates that T-duality should impose more than the 435 relations that hold in the minimal basis. Consequently, it is more legitimate to consider a basis that includes a larger set of independent couplings. The maximal basis, which has the greatest number of independent couplings, is therefore a more appropriate starting point for the analysis.

T-duality should fix the unambiguous couplings uniquely and should fix the ambiguous couplings up to some parameters that are removable by field redefinition. So it is more convenient to write the couplings in the maximal basis (5) as unambiguous and ambiguous terms. That is:

$$
\mathbf{S_1}^{(2)} = -\frac{2\alpha'^2}{\kappa^2} \int d^{10}x \sqrt{-G} e^{-2\Phi} \Big[ c_2 F_\alpha{}^{\gamma kl} F^{\alpha\beta ij} F_\beta{}^\delta{}_k{}^m F_\gamma{}^\epsilon{}_m{}^n F_\delta{}^\varepsilon{}_{ln} F_{\epsilon\varepsilon ij}
$$

$$
+ c_6 F_\alpha{}^\gamma{}_i{}^k F^{\alpha\beta ij} F_\beta{}^{\delta lm} F_\gamma{}^\epsilon{}_l{}^n F_\delta{}^\varepsilon{}_{kn} F_{\epsilon\varepsilon jm} + c_7 F_\alpha{}^{\gamma kl} F^{\alpha\beta ij} F_\beta{}^\delta{}_k{}^m F_\gamma{}^\epsilon{}_i{}^n F_\delta{}^\varepsilon{}_{ln} F_{\epsilon\varepsilon jm}
$$

$$
+ c_8 F_\alpha{}^\gamma{}_i{}^k F^{\alpha\beta ij} F_\beta{}^{\delta lm} F_\gamma{}^\epsilon{}_l{}^n F_\delta{}^\varepsilon{}_{km} F_{\epsilon\varepsilon jn} + c_{10} F_\alpha{}^\gamma{}_i{}^k F^{\alpha\beta ij} F_\beta{}^\delta{}_k{}^l F_\gamma{}^{\epsilon mn} F_\delta{}^\varepsilon{}_{lm} F_{\epsilon\varepsilon jn}
$$

$$
+ c_{11} F_{\alpha\beta}{}^{kl} F^{\alpha\beta ij} F_\gamma{}^\epsilon{}_{km} F^{\gamma\delta}{}_i{}^m F_\delta{}^\varepsilon{}_l{}^n F_{\epsilon\varepsilon jn} + c_{12} F_\alpha{}^\gamma{}_i{}^k F^{\alpha\beta ij} F_\beta{}^\delta{}_k{}^l F_\gamma{}^\epsilon{}_l{}^m F_\delta{}^\varepsilon{}_m{}^n F_{\epsilon\varepsilon jn}
$$

$$
+ c_{18} F_\alpha{}^\gamma{}_i{}^k F^{\alpha\beta ij} F_\beta{}^{\delta lm} F_\gamma{}^\epsilon{}_l{}^n F_\delta{}^\varepsilon{}_{jn} F_{\epsilon\varepsilon km} + c_{20} F_\alpha{}^\gamma{}_i{}^k F^{\alpha\beta ij} F_\beta{}^\delta{}_j{}^l F_\gamma{}^\epsilon{}_l{}^m F_\delta{}^\varepsilon{}_m{}^n F_{\epsilon\varepsilon kn}
$$

$$
+ c_{21} F_{\alpha\beta i}{}^k F^{\alpha\beta ij} F_\gamma{}^\epsilon{}_l{}^m F^{\gamma\delta}{}_j{}^l F_\delta{}^\varepsilon{}_m{}^n F_{\epsilon\varepsilon kn} + c_{22} F_\alpha{}^{\gamma kl} F^{\alpha\beta ij} F_\beta{}^\delta{}_k{}^m F_\gamma{}^\epsilon{}_i{}^n F_\delta{}^\varepsilon{}_{jn} F_{\epsilon\varepsilon lm}
$$

$$
+ c_{23} F_{\alpha\beta}{}^{kl} F^{\alpha\beta ij} F_\gamma{}^\epsilon{}_j{}^n F^{\gamma\delta}{}_i{}^m F_\delta{}^\varepsilon{}_{kn} F_{\epsilon\varepsilon lm} + c_{25} F_\alpha{}^\gamma{}_i{}^k F^{\alpha\beta ij} F_\beta{}^\delta{}_k{}^l F_\gamma{}^{\epsilon mn} F_\delta{}^\varepsilon{}_{jm} F_{\epsilon\varepsilon ln}
$$

$$
+ c_{27} F_{\alpha\beta}{}^{kl} F^{\alpha\beta ij} F_\gamma{}^\epsilon{}_j{}^n F^{\gamma\delta}{}_i{}^m F_\delta{}^\varepsilon{}_{km} F_{\epsilon\varepsilon ln} + c_{28} F_{\alpha\beta}{}^{kl} F^{\alpha\beta ij} F_\gamma{}^\epsilon{}_{jm} F^{\gamma\delta}{}_i{}^m F_\delta{}^\varepsilon{}_k{}^n F_{\epsilon\varepsilon ln}
$$

$$
+ c_{29} F_\alpha{}^\gamma{}_i{}^k F^{\alpha\beta ij} F_\beta{}^\delta{}_k{}^l F_\gamma{}^\epsilon{}_j{}^m F_\delta{}^\varepsilon{}_m{}^n F_{\epsilon\varepsilon ln} + c_{30} F_\alpha{}^\gamma{}_i{}^k F^{\alpha\beta ij} F_\beta{}^\delta{}_j{}^l F_\gamma{}^\epsilon{}_k{}^m F_\delta{}^\varepsilon{}_m{}^n F_{\epsilon\varepsilon ln}
$$

$$
+ c_{33} F_{\alpha\beta}{}^{kl} F^{\alpha\beta ij} F_\gamma{}^\epsilon{}_j{}^m F^{\gamma\delta}{}_{ik} F_\delta{}^\varepsilon{}_m{}^n F_{\epsilon\varepsilon ln} + c_{34} F_{\alpha\beta i}{}^k F^{\alpha\beta ij} F_\gamma{}^\epsilon{}_k{}^m F^{\gamma\delta}{}_j{}^l F_\delta{}^\varepsilon{}_m{}^n F_{\epsilon\varepsilon ln}
$$

$$
+ c_{36} F_\alpha{}^{\gamma kl} F^{\alpha\beta ij} F_{\beta\gamma}{}^{mn} F_\delta{}^\varepsilon{}_{jm} F^{\delta\epsilon}{}_{ik} F_{\epsilon\varepsilon ln} + c_{37} F_\alpha{}^\gamma{}_i{}^k F^{\alpha\beta ij} F_\beta{}^{\delta lm} F_\gamma{}^\epsilon{}_l{}^n F_\delta{}^\varepsilon{}_{jk} F_{\epsilon\varepsilon mn}
$$

$$
+ c_{39} F_\alpha{}^\gamma{}_i{}^k F^{\alpha\beta ij} F_\beta{}^\delta{}_j{}^l F_\gamma{}^\epsilon{}_l{}^m F_\delta{}^\varepsilon{}_k{}^n F_{\epsilon\varepsilon mn} + c_{40} F_\alpha{}^\gamma{}_i{}^k F^{\alpha\beta ij} F_\beta{}^\delta{}_j{}^l F_\gamma{}^\epsilon{}_k{}^m F_\delta{}^\varepsilon{}_l{}^n F_{\epsilon\varepsilon mn}
$$

$$
+ c_{42} F_\alpha{}^\gamma{}_i{}^k F^{\alpha\beta ij} F_\beta{}^\delta{}_{jk} F_\gamma{}^{\epsilon lm} F_\delta{}^\varepsilon{}_l{}^n F_{\epsilon\varepsilon mn} + c_{44} F_{\alpha\beta}{}^{kl} F^{\alpha\beta ij} F_\gamma{}^\epsilon{}_j{}^m F^{\gamma\delta}{}_{ik} F_\delta{}^\varepsilon{}_l{}^n F_{\epsilon\varepsilon mn}
$$

$$
+ c_{45} F_{\alpha\beta i}{}^k F^{\alpha\beta ij} F_\gamma{}^\epsilon{}_k{}^m F^{\gamma\delta}{}_j{}^l F_\delta{}^\varepsilon{}_l{}^n F_{\epsilon\varepsilon mn} + c_{53} F_\alpha{}^\gamma{}_i{}^k F^{\alpha\beta ij} F_{\beta\gamma}{}^m F_\delta{}^\varepsilon{}_l{}^n F^{\delta\epsilon}{}_{jk} F_{\epsilon\varepsilon mn}
$$

$$
+ c_{54} F_\alpha{}^\gamma{}_i{}^k F^{\alpha\beta ij} F_{\beta\gamma}{}^{lm} F_\delta{}^\varepsilon{}_k{}^n F^{\delta\epsilon}{}_{jl} F_{\epsilon\varepsilon mn} + c_{55} F_\alpha{}^\gamma{}_i{}^k F^{\alpha\beta ij} F_{\beta\gamma j}{}^l F_\delta{}^\varepsilon{}_l{}^n F^{\delta\epsilon}{}_k{}^m F_{\epsilon\varepsilon mn}
$$

$$
+ c_{56} F_\alpha{}^\gamma{}_i{}^k F^{\alpha\beta ij} F_{\beta\gamma jk} F_\delta{}^\varepsilon{}_l{}^n F^{\delta\epsilon lm} F_{\epsilon\varepsilon mn} + c_{57} F_{\alpha\beta}{}^{kl} F^{\alpha\beta ij} F_{\gamma\delta}{}^{mn} F^{\gamma\delta}{}_{ik} F_{\epsilon\varepsilon ln} F^{\epsilon\varepsilon}{}_{jm}
$$

$$
+ c_{58} F_{\alpha\beta}{}^{kl} F^{\alpha\beta ij} F_{\gamma\delta k}{}^n F^{\gamma\delta}{}_i{}^m F_{\epsilon\varepsilon ln} F^{\epsilon\varepsilon}{}_{jm} + c_{59} F_{\alpha\beta}{}^{kl} F^{\alpha\beta ij} F_{\gamma\delta k}{}^n F^{\gamma\delta}{}_i{}^m F_{\epsilon\varepsilon lm} F^{\epsilon\varepsilon}{}_{jn}
$$

$$
+ c_{63} F_{\alpha\beta i}{}^k F^{\alpha\beta ij} F_{\gamma\delta}{}^{mn} F^{\gamma\delta}{}_j{}^l F_{\epsilon\varepsilon ln} F^{\epsilon\varepsilon}{}_{km} + c_{65} F_{\alpha\beta i}{}^k F^{\alpha\beta ij} F_{\gamma\delta l}{}^m F^{\gamma\delta}{}_j{}^l F_{\epsilon\varepsilon mn} F^{\epsilon\varepsilon}{}_k{}^n
$$

$$
+ c_{66} F_{\alpha\beta i}{}^k F^{\alpha\beta ij} F_{\gamma\delta k}{}^m F^{\gamma\delta}{}_j{}^l F_{\epsilon\varepsilon mn} F^{\epsilon\varepsilon}{}_l{}^n + c_{71} F_\alpha{}^{\gamma kl} F^{\alpha\beta ij} F^{\delta\epsilon}{}_{ik} F^{\epsilon\mu}{}_{jl} H_{\beta\delta\varepsilon} H_{\gamma\epsilon\mu}
$$

$$
+ c_{73} F_\alpha{}^\gamma{}_i{}^k F^{\alpha\beta ij} F^{\delta\epsilon}{}_j{}^l F^{\epsilon\mu}{}_{kl} H_{\beta\delta\varepsilon} H_{\gamma\epsilon\mu} + c_{75} F_\alpha{}^{\gamma kl} F^{\alpha\beta ij} F_\delta{}^\varepsilon{}_{kl} F^{\delta\epsilon}{}_{ij} H_{\beta\varepsilon}{}^\mu H_{\gamma\epsilon\mu}
$$

$$
+ c_{76} F_\alpha{}^\gamma{}_i{}^k F^{\alpha\beta ij} F_\delta{}^\varepsilon{}_{kl} F^{\delta\epsilon}{}_j{}^l H_{\beta\varepsilon}{}^\mu H_{\gamma\epsilon\mu} + c_{82} F_\alpha{}^{\gamma kl} F^{\alpha\beta ij} F_\delta{}^\varepsilon{}_{jl} F^{\delta\epsilon}{}_{ik} H_{\beta\epsilon}{}^\mu H_{\gamma\varepsilon\mu}
$$

$$
+ c_{83} F_\alpha{}^\gamma{}_i{}^k F^{\alpha\beta ij} F_\delta{}^\varepsilon{}_{kl} F^{\delta\epsilon}{}_j{}^l H_{\beta\epsilon}{}^\mu H_{\gamma\varepsilon\mu} + c_{86} F_\alpha{}^\gamma{}_i{}^k F^{\alpha\beta ij} F_\beta{}^\delta{}_k{}^l F^{\epsilon\varepsilon}{}_{jl} H_{\gamma\epsilon}{}^\mu H_{\delta\varepsilon\mu}
$$

$$
+ c_{87} F_{\alpha\beta}{}^{kl} F^{\alpha\beta ij} F^{\gamma\delta}{}_{ik} F^{\epsilon\varepsilon}{}_{jl} H_{\gamma\epsilon}{}^\mu H_{\delta\varepsilon\mu} + c_{88} F_\alpha{}^\gamma{}_i{}^k F^{\alpha\beta ij} F_\beta{}^\delta{}_j{}^l F^{\epsilon\varepsilon}{}_{kl} H_{\gamma\epsilon}{}^\mu H_{\delta\varepsilon\mu}
$$

$$
+ c_{91} F_{\alpha\beta i}{}^k F^{\alpha\beta ij} F^{\gamma\delta}{}_j{}^l F^{\epsilon\varepsilon}{}_{kl} H_{\gamma\epsilon}{}^\mu H_{\delta\varepsilon\mu} + c_{94} F^{\alpha\beta ij} F^{\gamma\delta}{}_{ij} H_\alpha{}^{\epsilon\varepsilon} H_{\beta\epsilon}{}^\mu H_{\gamma\varepsilon}{}^\zeta H_{\delta\mu\zeta}
$$

$$
+ c_{97} F_\alpha{}^{\gamma kl} F^{\alpha\beta ij} F_\delta{}^\varepsilon{}_{kl} F^{\delta\epsilon}{}_{ij} H_{\beta\gamma}{}^\mu H_{\epsilon\varepsilon\mu} + c_{98} F_\alpha{}^\gamma{}_i{}^k F^{\alpha\beta ij} F_\delta{}^\varepsilon{}_{kl} F^{\delta\epsilon}{}_j{}^l H_{\beta\gamma}{}^\mu H_{\epsilon\varepsilon\mu}
$$

$$
+ c_{117} F^{\alpha\beta ij} F^{\gamma\delta}{}_{ij} H_{\alpha\gamma}{}^\epsilon H_\beta{}^{\varepsilon\mu} H_{\delta\varepsilon}{}^\zeta H_{\epsilon\mu\zeta} + c_{135} H_\alpha{}^{\delta\epsilon} H^{\alpha\beta\gamma} H_{\beta\delta}{}^\varepsilon H_\gamma{}^{\mu\zeta} H_{\epsilon\mu}{}^\eta H_{\varepsilon\zeta\eta}
$$

$$
+ c_{196} F_\alpha{}^{\gamma kl} F^{\alpha\beta ij} F_\delta{}^\varepsilon{}_{kl} F^{\delta\epsilon}{}_{ij} R_{\beta\gamma\epsilon\varepsilon} + c_{197} F_\alpha{}^\gamma{}_i{}^k F^{\alpha\beta ij} F_\delta{}^\varepsilon{}_{kl} F^{\delta\epsilon}{}_j{}^l R_{\beta\gamma\epsilon\varepsilon}
$$

$$
+ c_{200} F_\alpha{}^{\gamma kl} F^{\alpha\beta ij} F_\delta{}^\varepsilon{}_{kl} F^{\delta\epsilon}{}_{ij} R_{\beta\epsilon\gamma\varepsilon} + c_{201} F_\alpha{}^{\gamma kl} F^{\alpha\beta ij} F_\delta{}^\varepsilon{}_{jl} F^{\delta\epsilon}{}_{ik} R_{\beta\epsilon\gamma\varepsilon}
$$

$$
+ c_{202} F_\alpha{}^\gamma{}_i{}^k F^{\alpha\beta ij} F_\delta{}^\varepsilon{}_{kl} F^{\delta\epsilon}{}_j{}^l R_{\beta\epsilon\gamma\varepsilon} + c_{205} R_\alpha{}^\epsilon{}_\gamma{}^\varepsilon R^{\alpha\beta\gamma\delta} R_{\beta\epsilon\delta\varepsilon}
$$

$$
+ c_{207} F_{\alpha\beta}{}^{kl} F^{\alpha\beta ij} F^{\gamma\delta}{}_{ik} F^{\epsilon\varepsilon}{}_{jl} R_{\gamma\delta\epsilon\varepsilon} + c_{208} F_{\alpha\beta i}{}^k F^{\alpha\beta ij} F^{\gamma\delta}{}_j{}^l F^{\epsilon\varepsilon}{}_{kl} R_{\gamma\delta\epsilon\varepsilon}
$$

$$
+ c_{210} F_\alpha{}^\gamma{}_i{}^k F^{\alpha\beta ij} F_\beta{}^\delta{}_k{}^l F^{\epsilon\varepsilon}{}_{jl} R_{\gamma\epsilon\delta\varepsilon} + c_{211} F_\alpha{}^\gamma{}_i{}^k F^{\alpha\beta ij} F_\beta{}^\delta{}_j{}^l F^{\epsilon\varepsilon}{}_{kl} R_{\gamma\epsilon\delta\varepsilon}
$$

$$+c_{213}R_{\alpha\beta}{}^{\epsilon\varepsilon}R^{\alpha\beta\gamma\delta}R_{\gamma\epsilon\delta\varepsilon} + c_{259}F_\alpha{}^{\beta ij}F_\gamma{}^\epsilon{}_k{}^l F^{\gamma\delta}{}_i{}^k F_\epsilon{}^\varepsilon{}_{jl}H_{\beta\delta\varepsilon}\nabla^\alpha\Phi$$

$$+c_{262}F_\alpha{}^{\beta ij}F_\gamma{}^\epsilon{}_j{}^l F^{\gamma\delta}{}_i{}^k F_\delta{}^\varepsilon{}_{kl}H_{\beta\epsilon\varepsilon}\nabla^\alpha\Phi + c_{266}F_\alpha{}^{\beta ij}F_\beta{}^\gamma{}_i{}^k F_\delta{}^\varepsilon{}_{kl}F^{\delta\epsilon}{}_j{}^l H_{\gamma\epsilon\varepsilon}\nabla^\alpha\Phi$$

$$+c_{327}F_\alpha{}^{\gamma kl}F^{\alpha\beta ij}F^{\delta\epsilon}{}_{ik}H_{\gamma\epsilon\varepsilon}\nabla_\beta F_\delta{}^\varepsilon{}_{jl} + c_{356}F_\alpha{}^{\gamma kl}F^{\alpha\beta ij}F^{\delta\epsilon}{}_{ij}\nabla_\beta\nabla_\epsilon F_{\gamma\delta kl}$$

$$+c_{357}F_\alpha{}^\gamma{}_i{}^k F^{\alpha\beta ij}F^{\delta\epsilon}{}_j{}^l\nabla_\beta\nabla_\epsilon F_{\gamma\delta kl} + c_{363}F_\alpha{}^{\gamma ij}F_\beta{}^{\delta kl}F_\gamma{}^\epsilon{}_{kl}F_{\delta\epsilon ij}\nabla^\alpha\Phi\nabla^\beta\Phi$$

$$+c_{364}F_\alpha{}^{\gamma ij}F_\beta{}^{\delta kl}F_\gamma{}^\epsilon{}_{ik}F_{\delta\epsilon jl}\nabla^\alpha\Phi\nabla^\beta\Phi + c_{365}F_\alpha{}^{\gamma ij}F_\beta{}^\delta{}_i{}^k F_\gamma{}^\epsilon{}_k{}^l F_{\delta\epsilon jl}\nabla^\alpha\Phi\nabla^\beta\Phi$$

$$+c_{367}F_\alpha{}^{\gamma ij}F_\beta{}^\delta{}_i{}^k F_\gamma{}^\epsilon{}_j{}^l F_{\delta\epsilon kl}\nabla^\alpha\Phi\nabla^\beta\Phi + c_{369}F_\alpha{}^{\gamma ij}F_{\beta\gamma}{}^{kl}F_{\delta\epsilon jl}F^{\delta\epsilon}{}_{ik}\nabla^\alpha\Phi\nabla^\beta\Phi$$

$$+c_{370}F_\alpha{}^{\gamma ij}F_{\beta\gamma i}{}^k F_{\delta\epsilon kl}F^{\delta\epsilon}{}_j{}^l\nabla^\alpha\Phi\nabla^\beta\Phi + c_{380}H_\alpha{}^{\gamma\delta}H_\beta{}^{\epsilon\varepsilon}H_{\gamma\epsilon}{}^\mu H_{\delta\varepsilon\mu}\nabla^\alpha\Phi\nabla^\beta\Phi$$

$$+c_{455}F_\alpha{}^{\gamma kl}F^{\alpha\beta ij}\nabla_\beta F^{\delta\epsilon}{}_{ik}\nabla_\gamma F_{\delta\epsilon jl} + c_{456}F_\alpha{}^\gamma{}_i{}^k F^{\alpha\beta ij}\nabla_\beta F^{\delta\epsilon}{}_j{}^l\nabla_\gamma F_{\delta\epsilon kl}$$

$$+c_{460}F_\alpha{}^{\gamma kl}F^{\alpha\beta ij}\nabla_\beta F_{\delta\epsilon kl}\nabla_\gamma F^{\delta\epsilon}{}_{ij} + c_{462}F_\alpha{}^\gamma{}_i{}^k F^{\alpha\beta ij}\nabla_\beta F_{\delta\epsilon kl}\nabla_\gamma F^{\delta\epsilon}{}_j{}^l$$

$$+c_{495}F_\alpha{}^{\gamma kl}F^{\alpha\beta ij}F^{\delta\epsilon}{}_{ik}\nabla_\gamma\nabla_\epsilon F_{\beta\delta jl} + c_{723}F^{\alpha\beta ij}F^{\gamma\delta kl}\nabla_\delta F_{\beta\epsilon jl}\nabla^\epsilon F_{\alpha\gamma ik}$$

$$+c_{724}F^{\alpha\beta ij}F^{\gamma\delta}{}_i{}^k\nabla_\beta F_{\delta\epsilon kl}\nabla^\epsilon F_{\alpha\gamma j}{}^l + c_{766}H_\alpha{}^{\delta\epsilon}H^{\alpha\beta\gamma}H_{\beta\delta}{}^\varepsilon H_\gamma{}^{\mu\zeta}\nabla_\varepsilon H_{\epsilon\mu\zeta} + \cdots \Big], \qquad (7)$$

where dots represent 718 ambiguous couplings. The above basis includes the YM field strength and its derivatives, such as $\text{Tr}(FF)$, $\text{Tr}(FFF)$, $\text{Tr}(FFFF)$, and $\text{Tr}(FFFFFF)$. In the above equation, $c_1, c_2, \cdots, c_{801}$ are 801 background-independent coupling constants that will be found in the next section using T-duality. We will find that T-duality fixes all unambiguous couplings which have traces of more than two $F$'s and their derivatives to be zero. In other words, the unambiguous couplings involving $\text{Tr}(FFF)$, $\text{Tr}(FFFF)$, $\text{Tr}(FFFFFF)$, and their derivatives, are set to zero by the T-duality constraint.

There are two other sets of couplings at 6-derivative order with fixed coupling constants that result from replacing $H \to H - (3\alpha'/2)\Omega$ into the 2- and 4-derivative orders. This replacement into the 2-derivative order (see eq.(15)) produces the following coupling at 6-derivative order:

$$\mathbf{S_2}^{(2)} = -\frac{2\alpha'^2}{\kappa^2}\int d^{10}x\sqrt{-G}\,e^{-2\Phi}\Big[-\frac{3}{16}\Omega_{\mu\nu\alpha}\Omega^{\mu\nu\alpha}\Big]. \qquad (8)$$

The Chern-Simons three-form is given by:

$$\Omega_{\mu\nu\alpha} = \omega_{[\mu\mu_1}{}^{\nu_1}\partial_\nu\omega_{\alpha]\nu_1}{}^{\mu_1} + \frac{2}{3}\omega_{[\mu\mu_1}{}^{\nu_1}\omega_{\nu\nu_1}{}^{\alpha_1}\omega_{\alpha]\alpha_1}{}^{\mu_1}\;;\;\; \omega_{\mu\mu_1}{}^{\nu_1} = e^\nu{}_{\mu_1}\nabla_\mu e_\nu{}^{\nu_1}, \qquad (9)$$

where $e_\mu{}^{\mu_1}e_\nu{}^{\nu_1}\eta_{\mu_1\nu_1} = G_{\mu\nu}$. The covariant derivative in the definition of the spin connection applies only on the curved indices of the frame $e_\mu{}^{\mu_1}$. Our index convention is that $\mu, \nu, \ldots$ are the indices of the curved spacetime, and $\mu_1, \nu_1, \ldots$ are the indices of the flat tangent space.

The action at the 4-derivative order depends on the scheme. We consider the 4-derivative couplings in the Meissner scheme, in which the YM couplings are added [5] (see eq.(17)). The nice property of this action is that these couplings do not change the propagators that are produced by the couplings at the 2-derivative order. This action also manifestly satisfies the T-duality constraint at the 4-derivative order [5]. The Green-Schwarz replacement $H \to H - (3\alpha'/2)\Omega$ into this action produces the following couplings at the 6-derivative order:

$$\mathbf{S_3}^{(2)} = -\frac{2\alpha'^2}{\kappa^2}\int d^{10}x\sqrt{-G}\,e^{-2\Phi}\Big[\frac{3}{32}H_{\gamma\delta}{}^\epsilon R^{\alpha\beta\gamma\delta}\Omega_{\alpha\beta\epsilon} - \frac{3}{32}F^{\alpha\beta ij}F^{\gamma\delta}{}_{ij}H_{\alpha\gamma}{}^\epsilon\Omega_{\beta\delta\epsilon}$$

$$-\frac{3}{16}H_\gamma{}^{\delta\epsilon}R^{\alpha\beta}{}_\alpha{}^\gamma\Omega_{\beta\delta\epsilon} + \frac{3}{64}F^{\alpha\beta ij}F^{\gamma\delta}{}_{ij}H_{\alpha\beta}{}^\epsilon\Omega_{\gamma\delta\epsilon} + \frac{1}{16}H^{\gamma\delta\epsilon}R^{\alpha\beta}{}_{\alpha\beta}\Omega_{\gamma\delta\epsilon}$$

$$-\frac{3}{16}H_\beta{}^{\delta\epsilon}R^{\alpha\beta}{}_\alpha{}^\gamma\Omega_{\gamma\delta\epsilon} + \frac{3}{32}H_{\alpha\beta}{}^\epsilon R^{\alpha\beta\gamma\delta}\Omega_{\gamma\delta\epsilon} - \frac{1}{32}H_\alpha{}^{\delta\epsilon}H^{\alpha\beta\gamma}H_{\beta\delta}{}^\varepsilon\Omega_{\gamma\epsilon\varepsilon}$$

$$+\frac{3}{32}H_{\alpha\beta}{}^\delta H^{\alpha\beta\gamma}H_\gamma{}^{\epsilon\varepsilon}\Omega_{\delta\epsilon\varepsilon} - \frac{1}{192}H_{\alpha\beta\gamma}H^{\alpha\beta\gamma}H^{\delta\epsilon\varepsilon}\Omega_{\delta\epsilon\varepsilon} + \frac{1}{4}H^{\beta\gamma\delta}\Omega_{\beta\gamma\delta}\nabla_\alpha\nabla^\alpha\Phi$$

$$-\frac{1}{4}H^{\beta\gamma\delta}\Omega_{\beta\gamma\delta}\nabla_\alpha\Phi\nabla^\alpha\Phi - \frac{3}{4}H_\alpha{}^{\gamma\delta}\Omega_{\beta\gamma\delta}\nabla^\beta\nabla^\alpha\Phi \bigg]. \tag{10}$$

The effective action at the 6-derivative order then is:

$$\mathbf{S}^{(2)} \;\; = \;\; \mathbf{S_1}^{(2)} + \mathbf{S_2}^{(2)} + \mathbf{S_3}^{(2)}. \tag{11}$$

In the next section, we impose T-duality on this action to find the couplings in the maximal basis $\mathbf{S_1}^{(2)}$.

# 3 T-duality Constraint on the Maximal Basis

Having found the maximal basis, we now impose the T-duality on the circular reduction of the couplings to find the corresponding coupling constants. The circular reduction of the couplings and the corresponding T-duality transformations involve the scalar component of the YM fields nonlinearly [21]. It has been proposed in [5] that the imposition of the truncated T-duality transformations on the truncated reduction of the couplings has enough information to fix the coupling constants. This constraint is the following [5]:

$$\sum_{n=0}^\infty \alpha'^n \left[ S^{L(n)}(\psi) - \sum_{m=0}^\infty \alpha'^m [S^{(n,m)}(\psi_0^L)]^L - \int d^9x\, \partial^a \left[ e^{-2\bar\phi} J_a^{L(n+1/2)}(\psi) \right] \right] = 0, \tag{12}$$

where the superscript $L$ in each term indicates that only the zeroth and first order terms of the scalar should be retained. We refer the interested readers to [5] for details of each term above. It has been observed in [4] that the odd-derivative couplings in the effective action and in the corrections to the Buscher rules are zero, hence, $m$ and $n$ in the above equation take only integer values.

The Taylor expansion of the $\alpha'^n$-order action $S^n$ at order $\alpha'^m$ has the following contributions:

$$S^{(n,m)} \;\; = \;\; \sum_{p=\{m\}} S_{(p)}^{(n,m)}, \tag{13}$$

where $\{m\}$ is the number of partitions of $m$, e.g., $\{2\} = \{(1,1),(2)\}$. $(1,1)$ represents two first-order corrections to the Buscher rules, and $(2)$ represents one second-order correction to the Buscher rules. Using this relation, one may write (12) as:

$$\sum_{n=0}^\infty \alpha'^n \left[ -\sum_{m=1}^\infty \alpha'^m [S_{(m)}^{(n,m)}(\psi_0^L)]^L - \int d^9x\, \partial^a \left[ e^{-2\bar\phi} J_a^{L(n+1/2)}(\psi) \right] \right]$$

$$= \sum_{n=0}^{\infty} \alpha'^n \left[ S^{L(n)}(\psi_0^L) - S^{L(n)}(\psi) + \sum_{m=1}^{\infty} \sum_{p'=\{m'\}} \alpha'^m [S_{(p')}^{(n,m)}(\psi_0^L)]^L \right], \quad (14)$$

where $\{m'\}$ is the number of partitions of $m$ that do not use $m$, e.g., $\{2'\} = \{(1,1)\}$.

At a given order of $\alpha'$, the terms on the left-hand side of (14) have arbitrary parameters in the total derivative terms and in the correction to the Buscher rules at that order of $\alpha'$, whereas the terms on the right-hand side have arbitrary coupling constants at that order of $\alpha'$ and all other terms from the Taylor expansion in the last term are fixed at the lower orders of $\alpha'$. This equation then has a homogeneous solution which satisfies the homogeneous part of the equation (14) where the right-hand side is zero. We are not interested in this homogeneous solution. Instead, we are interested in the particular parameters that satisfy the inhomogeneous equation, where the right-hand side is not zero. The particular solution should fix the parameters in terms of the coupling constants in the maximal basis and the fixed numbers on the right-hand side of (14). It should also fix some relations between the coupling constants and the fixed numbers.

To determine the appropriate constraints on the effective actions, terms at every order of $\alpha'$ must be equated on the two sides of (14). Using the reduction scheme for the NS-NS and YM fields [20, 21], the above constraint at order $\alpha'^0$ has been used to fix the effective action to be [5]

$$\mathbf{S}^{(0)} = -\frac{2}{\kappa^2} \int d^{10}x \sqrt{-G} e^{-2\Phi} \left[ R - \frac{1}{12} H_{\alpha\beta\gamma} H^{\alpha\beta\gamma} + 4\nabla_\alpha \Phi \nabla^\alpha \Phi - \frac{1}{4} F_{\mu\nu ij} F^{\mu\nu ij} \right], \quad (15)$$

which is the bosonic part of the standard effective action of heterotic theory [19, 15]. The corresponding truncated Buscher rules $\psi_0^L$ have been found to be

$$\varphi^L = -\varphi \,, \quad g_a^L = b_a \,, \quad b_a^L = g_a \,, \quad \bar{g}_{ab}^L = \bar{g}_{ab} \,, \quad (16)$$
$$\bar{H}_{abc}^L = \bar{H}_{abc} \,, \quad \bar{\phi}^L = \bar{\phi} \,, (\bar{A}_a^L)^{ij} = \bar{A}_a^{\,ij} \,, \quad (\alpha^L)^{ij} = -\alpha^{ij} \,,$$

where the base space fields are defined in the reduction of the NS-NS and YM fields with the notation that has been used in [5].

This constraint (14) at order $\alpha'$ has been used in [5] to find both the effective action at 4-derivative order and the corrections to the truncated Buscher rules (16) at 2-derivative order. The couplings in the Meissner scheme are found to be the following [5]:

$$\mathbf{S}^{(1)} = -\frac{2\alpha'}{8\kappa^2} \int d^{10}x \sqrt{-G} e^{-2\Phi} \left[ \frac{1}{4} F_\alpha{}^{\gamma kl} F^{\alpha\beta ij} F_\beta{}^\delta{}_{kl} F_{\gamma\delta ij} + \frac{1}{2} F_\alpha{}^\gamma{}_{ij} F^{\alpha\beta ij} F_\beta{}^{\delta kl} F_{\gamma\delta kl} \right.$$
$$- \frac{1}{8} F_{\alpha\beta}{}^{kl} F^{\alpha\beta ij} F_{\gamma\delta kl} F^{\gamma\delta}{}_{ij} - \frac{1}{16} F_{\alpha\beta ij} F^{\alpha\beta ij} F_{\gamma\delta kl} F^{\gamma\delta kl} + \frac{1}{4} F^{\alpha\beta ij} F^{\gamma\delta}{}_{ij} H_{\alpha\gamma}{}^\epsilon H_{\beta\delta\epsilon}$$
$$- \frac{1}{8} F^{\alpha\beta ij} F^{\gamma\delta}{}_{ij} H_{\alpha\beta}{}^\epsilon H_{\gamma\delta\epsilon} + \frac{1}{24} H_\alpha{}^{\delta\epsilon} H^{\alpha\beta\gamma} H_{\beta\delta}{}^\varepsilon H_{\gamma\epsilon\varepsilon} - \frac{1}{8} H_{\alpha\beta}{}^\delta H^{\alpha\beta\gamma} H_\gamma{}^{\epsilon\varepsilon} H_{\delta\epsilon\varepsilon}$$
$$+ \frac{1}{144} H_{\alpha\beta\gamma} H^{\alpha\beta\gamma} H_{\delta\epsilon\varepsilon} H^{\delta\epsilon\varepsilon} + H_\alpha{}^{\gamma\delta} H_{\beta\gamma\delta} R^{\alpha\beta} - 4R_{\alpha\beta} R^{\alpha\beta} - \frac{1}{6} H_{\alpha\beta\gamma} H^{\alpha\beta\gamma} R + R^2$$
$$+ R_{\alpha\beta\gamma\delta} R^{\alpha\beta\gamma\delta} - \frac{1}{2} H_\alpha{}^{\delta\epsilon} H^{\alpha\beta\gamma} R_{\beta\gamma\delta\epsilon} - \frac{2}{3} H_{\beta\gamma\delta} H^{\beta\gamma\delta} \nabla_\alpha \nabla^\alpha \Phi + \frac{2}{3} H_{\beta\gamma\delta} H^{\beta\gamma\delta} \nabla_\alpha \Phi \nabla^\alpha \Phi$$

$$+8R\nabla_\alpha\Phi\nabla^\alpha\Phi - 16R_{\alpha\beta}\nabla^\alpha\Phi\nabla^\beta\Phi + 16\nabla_\alpha\Phi\nabla^\alpha\Phi\nabla_\beta\Phi\nabla^\beta\Phi - 32\nabla^\alpha\Phi\nabla_\beta\Phi\nabla_\alpha\Phi\nabla^\beta\Phi$$

$$+2H_\alpha{}^{\gamma\delta}H_{\beta\gamma\delta}\nabla^\beta\nabla^\alpha\Phi + 2H^{\alpha\beta\gamma}\Omega_{\alpha\beta\gamma}\Big] \,. \tag{17}$$

The corresponding 2-derivative corrections to the truncated Buscher rules (16) are [5]:

$$-8\Delta\bar\phi^{(1)} = -\frac{1}{2}e^{\varphi/2}\bar F_{abij}V^{ab}\alpha^{ij} - \frac{1}{2}e^{-\varphi/2}\bar F_{abij}W^{ab}\alpha^{ij} \,, \tag{18}$$

$$-8\Delta\bar g_{ab}^{(1)} = -4e^{\varphi/2}\bar F_{\{b}{}^{cij}V_{a\}c}\alpha_{ij} - 4e^{-\varphi/2}\bar F_{\{b}{}^{cij}W_{a\}c}\alpha_{ij} \,,$$

$$-8\Delta\alpha_{ij}^{(1)} = -e^\varphi V_{cd}V^{cd}\alpha_{ij} + e^{-\varphi}W_{cd}W^{cd}\alpha_{ij} + 2\alpha_{ij}\partial_c\partial^c\varphi - 4\alpha_{ij}\partial_c\varphi\partial^c\bar\phi \,,$$

$$-8\Delta\bar A_a^{(1)}{}_{ij} = 0 \,,$$

$$-8\Delta B_{ab}^{(1)} = 4V_{[b}{}^cW_{a]c} + 2e^{\varphi/2}\bar F_{[b}{}^{cij}V_{a]c}\alpha_{ij} + 2e^{-\varphi/2}\bar F_{[b}{}^{cij}W_{a]c}\alpha_{ij} \,,$$

$$-8\Delta\varphi^{(1)} = -e^\varphi V_{ab}V^{ab} - e^{-\varphi}W_{ab}W^{ab} - 2\partial_a\varphi\partial^a\varphi + 2V^{ab}W_{ab} \,,$$

$$-8\Delta g_a^{(1)} = -e^{\varphi/2}\bar H_{abc}V^{bc} - 2e^{-\varphi/2}W_{ab}\partial^b\varphi + \frac{1}{2}e^{-\varphi/2}\bar H_{abc}W^{bc} - e^{\varphi/2}V_{ab}\partial^b\varphi - \frac{1}{2}\bar H_{abc}\bar F^{bcij}\alpha_{ij} \,,$$

and $\Delta b_a^{(1)}(\psi) = -\Delta g_a^{(1)}(\psi_0^L)$. They are added to the truncated Buscher rules (16) as:

$$\varphi^L = -\varphi + \alpha'\Delta\varphi^{(1)} \,, \quad g_a^L = b_a + \alpha' e^{\varphi/2}\Delta g_a^{(1)} \,,$$
$$b_a^L = g_a + \alpha' e^{-\varphi/2}\Delta b_a^{(1)} \,, \quad \bar g_{ab}^L = \bar g_{ab} + \alpha'\Delta\bar g_{ab}^{(1)} \,,$$
$$\bar H_{abc}^L = \bar H_{abc} + \alpha'\Delta\bar H_{abc}^{(1)} \,, \quad \bar\phi^L = \bar\phi + \alpha'\Delta\bar\phi^{(1)} \,, \tag{19}$$
$$(\bar A_a^L)^{ij} = \bar A_a{}^{ij} + \alpha'\Delta\bar A_a^{(1)ij} \,, \quad (\alpha^L)^{ij} = -\alpha^{ij} + \alpha'\Delta\alpha^{(1)ij} \,.$$

The correction $\Delta\bar H^{(1)}$ is related to the corrections $\Delta B^{(1)}$, $\Delta g^{(1)}$, $\Delta b^{(1)}$ and $\Delta\bar A_{ij}^{(1)}$ as:

$$\Delta\bar H_{abc}^{(1)} = 3\partial_{[a}\Delta B_{bc]}^{(1)} - 3e^{\varphi/2}V_{[ab}\Delta g_{c]}^{(1)} - 3e^{-\varphi/2}W_{[ab}\Delta b_{c]}^{(1)} - 3\bar F_{[ab}{}^{ij}\Delta\bar A_{c]ij}^{(1)} \,, \tag{20}$$

which results from the transformation of the $\bar H$-Bianchi identity in the base space under the T-duality transformation at order $\alpha'$.

## 3.1  T-duality at 6-derivative order

The constraint in (14) at order $\alpha'^2$ is:

$$-[S_{(2)}^{(0,2)}(\psi_0^L)]^L - \int d^9x\,\partial^a\left[e^{-2\bar\phi}J_a^{L(5/2)}(\psi)\right] = S^{L(2)}(\psi_0^L) - S^{L(2)}(\psi) + [S_{(1)}^{(1,1)}(\psi_0^L)]^L + [S_{(1,1)}^{(0,2)}(\psi_0^L)]^L. \tag{21}$$

To solve the above equation, one must assume that the total derivative term $J_a^{L(5/2)}(\psi)$ includes all contractions of the base space fields $\partial\varphi$, $\partial\bar\phi$, $e^{\varphi/2}V$, $e^{-\varphi/2}W$, $\bar H$, and $\bar F_{ab}{}^{ij}$ at the 5-derivative order, with arbitrary coefficients. Moreover, one needs to include all 4-derivative corrections to

the Buscher rules given in (19)

$$\varphi^L = -\varphi + \alpha'\Delta\varphi^{(1)} + \frac{\alpha'^2}{2}\Delta\varphi^{(2)}, \quad g_a^L = b_a + \alpha'e^{\varphi/2}\Delta g_a^{(1)} + \frac{\alpha'^2}{2}e^{\varphi/2}\Delta g_a^{(2)},$$

$$b_a^L = g_a + \alpha'e^{-\varphi/2}\Delta b_a^{(1)} + \frac{\alpha'^2}{2}e^{-\varphi/2}\Delta b_a^{(2)}, \quad \bar{g}_{ab}^L = \bar{g}_{ab} + \alpha'\Delta\bar{g}_{ab}^{(1)} + \frac{\alpha'^2}{2}\Delta\bar{g}_{ab}^{(2)}, \tag{22}$$

$$\bar{H}_{abc}^L = \bar{H}_{abc} + \alpha'\Delta\bar{H}_{abc}^{(1)} + \frac{\alpha'^2}{2}\Delta\bar{H}_{abc}^{(2)}, \quad \bar{\phi}^L = \bar{\phi} + \alpha'\Delta\bar{\phi}^{(1)} + \frac{\alpha'^2}{2}\Delta\bar{\phi}^{(2)},$$

$$(\bar{A}_a^L)^{ij} = \bar{A}_a{}^{ij} + \alpha'\Delta\bar{A}_a^{(1)ij} + \frac{\alpha'^2}{2}\Delta\bar{A}_a^{(2)ij}, \quad (\alpha^L)^{ij} = -\alpha^{ij} + \alpha'\Delta\alpha^{(1)ij} + \frac{\alpha'^2}{2}\Delta\alpha^{(2)ij}.$$

The relation between $\Delta\bar{H}^{(2)}$ and the corrections $\Delta B^{(2)}$, $\Delta g^{(2)}$, $\Delta b^{(2)}$, $\Delta\bar{A}_{ij}^{(2)}$, $\Delta g^{(1)}$, $\Delta b^{(1)}$, and $\Delta\bar{A}_{ij}^{(1)}$ results from the transformation of the $\bar{H}$-Bianchi identity under the T-duality transformation at order $\alpha'^2$, as:

$$\begin{aligned}\Delta\bar{H}_{abc}^{(2)} = {}& 3\partial_{[a}\Delta B_{bc]}^{(2)} - 3e^{\varphi/2}V_{[ab}\Delta g_{c]}^{(2)} - 3e^{-\varphi/2}W_{[ab}\Delta b_{c]}^{(2)} - 3\bar{F}_{[ab}{}^{ij}\Delta\bar{A}_{c]ij}^{(2)} \\ & -6\Delta\bar{A}_{[a}^{(1)ij}\partial_b\Delta\bar{A}_{c]ij}^{(1)} - 6\Delta\bar{g}_{[a}^{(1)}\partial_b\Delta\bar{b}_{c]}^{(1)} - 6\Delta\bar{b}_{[a}^{(1)}\partial_b\Delta\bar{g}_{c]}^{(1)} + 6\Delta\bar{b}_{[a}^{(1)}\partial_b\varphi\Delta\bar{g}_{c]}^{(1)}.\end{aligned} \tag{23}$$

The corrections $\Delta B_{ab}^{(2)}$, $\Delta\varphi^{(2)}$, $\Delta g_a^{(2)}$, $\Delta b_a^{(2)}$, $\Delta\bar{g}_{ab}^{(2)}$, $\Delta\bar{\phi}^{(2)}$, $\Delta\bar{A}_a^{(2)ij}$, $\Delta\alpha^{(2)ij}$ can be written as all contractions of the base space fields at the four-derivative order with arbitrary parameters for each contraction.

The circular reduction of the frame $e_\mu{}^{\mu_1}$ is given by:

$$e_\mu{}^{\mu_1} = \begin{pmatrix} \bar{e}_a{}^{a_1} & 0 \\ e^{\varphi/2}g_a & e^{\varphi/2} \end{pmatrix}, \tag{24}$$

where $\bar{e}_a{}^{a_1}\bar{e}_b{}^{b_1}\eta_{a_1b_1} = \bar{g}_{ab}$. Using this reduction and the other NS-NS and YM reductions, it is a straightforward calculation to determine the circular reduction of the 6-derivative couplings in (11) to calculate $S^{L(2)}(\psi)$, and its transformation under the leading order T-duality (16) to calculate $S^{L(2)}(\psi_0^L)$.

To calculate $[S_{(1)}^{(1,1)}(\psi_0^L)]^L$, one first needs to calculate the reduction of the 4-derivative couplings in (17) for curved base space, because the T-duality corrections at order $\alpha'$ in (18) have a non-zero $\Delta\bar{g}_{ab}^{(1)}$. Using the reduction (24) and the other NS-NS and YM reductions, one can calculate $S^{(1)}$. Since the correction $\Delta\alpha^{(1)ij}$ in (18) is proportional to the scalar $\alpha^{ij}$, one truncates $S^{(1)}$ to produce $S^{L(1)}(\psi)$. This reduced action includes, among other things, the spin connection $\bar{\omega}_{abc}$ of the base space, which results from the reduction of the last term in (17). In the Taylor expansion of $S^{L(1)}(\psi)$, the correction $\Delta\bar{\omega}_{abc}^{(1)}$ then appears. This correction is related to the metric correction as:

$$\Delta\bar{\omega}_{abc}^{(1)} = \frac{1}{2}\partial_c\Delta\bar{g}_{ab}^{(1)} - \frac{1}{2}\partial_b\Delta\bar{g}_{ac}^{(1)}, \tag{25}$$

where $\Delta\bar{g}_{ab}^{(1)}$ is given in (18). The above relation has been found from perturbing the following relation for the spin connection around flat space:

$$\bar{\omega}_{abc} = \frac{1}{2}\partial_c g_{ab} - \frac{1}{2}\partial_b g_{ac} + \frac{1}{2}\partial_a\bar{e}_b{}^{a_1}\bar{e}_{ca_1} - \frac{1}{2}\partial_a\bar{e}_c{}^{a_1}\bar{e}_{ba_1} \tag{26}$$

and used the fact that $2\Delta\bar{e}_a{}^{a_1}\bar{e}_{ba_1} = \Delta g_{ab}$. Note that $\bar{\omega}_{abc}$ is antisymmetric with respect to its last two indices. Then, using the correction (25) and the corrections in (18), one can Taylor expand $S^{L(1)}(\psi)$ around $\psi_0^L$ to calculate $[S^{(1,1)}(\psi_0^L)]^L$ in flat base space.

To calculate $[S_{(1,1)}^{(0,2)}(\psi_0^L)]^L$, we use the truncated reduction of the leading-order action $S^{L(0)}$, because the first-order correction $\Delta\alpha^{(1)ij}$ in (18) is proportional to the scalar $\alpha^{ij}$. Otherwise, one should consider the terms in $S^{(0)}$ that have second order of the scalar field as well. We then Taylor expand $S^{L(0)}$ and keep the terms that have two first-order corrections. There is also another contribution to $[S_{(1,1)}^{(0,2)}(\psi_0^L)]^L$ from the second-order correction $\Delta\bar{H}_{abc}^{(2)}$ that is replaced by the relation (23). In this way, one can calculate $[S_{(1,1)}^{(0,2)}(\psi_0^L)]^L$.

The calculation of $[S_{(2)}^{(0,2)}(\psi_0^L)]^L$ in terms of the second-order corrections is similar to the calculation of $[S^{(0,1)}(\psi_0^L)]^L$ in terms of the first-order correction that has been found in [5]. The only difference is that one should replace the first-order corrections in [5] with the second-order corrections.

The final step for solving the equation (21) is to impose the Bianchi identities associated with the field strengths $\bar{H}$, $\bar{F}$, $V$, and $W$. We impose the $\bar{H}$-Bianchi identity in its gauge-invariant form, while for the other Bianchi identities, we impose them in a non-gauge-invariant form [22]. The solution of the resulting system of linear algebraic equations provides an expression for the parameters of the second-order corrections to the Buscher rules. These parameters are expressed in terms of the coupling constants $c_1, c_2, \ldots, c_{801}$ and the fixed numbers resulting from the couplings at 2- and 4-derivative orders. This solution also establishes 468 relationships between the coupling constants and the fixed numbers. Replacing these 468 relations into the maximal basis (7), one finds that all unambiguous couplings in (7) become zero except 3 of them. That is,

$$\mathbf{S_1}^{(2)} = -\frac{2\alpha'^2}{\kappa^2}\int d^{10}x\sqrt{-G}e^{-2\Phi}\Big[-\frac{1}{64}F^{\alpha\beta ij}F^{\gamma\delta}{}_{ij}H_{\alpha\gamma}{}^{\epsilon}H_{\beta}{}^{\varepsilon\mu}H_{\delta\varepsilon}{}^{\zeta}H_{\epsilon\mu\zeta}$$
$$-\frac{1}{768}H_{\alpha}{}^{\delta\epsilon}H^{\alpha\beta\gamma}H_{\beta\delta}{}^{\varepsilon}H_{\gamma}{}^{\mu\zeta}H_{\epsilon\mu}{}^{\eta}H_{\varepsilon\zeta\eta} - \frac{1}{128}H_{\alpha}{}^{\delta\epsilon}H^{\alpha\beta\gamma}H_{\beta\delta}{}^{\varepsilon}H_{\gamma}{}^{\mu\zeta}\nabla_{\varepsilon}H_{\epsilon\mu\zeta} + \cdots\Big], \quad (27)$$

where the dots represent 717 ambiguous couplings. The coupling constants of these couplings have fixed numbers as well as 333 unfixed parameters. We have found that these parameters are removable by the freedom due to the field redefinitions, integration by parts, and the Bianchi identities. Hence, these parameters can be fixed to any arbitrary values. We have found that for no specific values for these parameters, the first derivatives of the Riemann, Ricci and scalar curvature, the second derivatives of the $H$-field and $F$-field, and the third derivatives of the dilaton become zero. For the case that all 333 parameters are zero, the ambiguous couplings in the above equation have 257 non-zero coupling constants.

Since the T-duality constraint produces 468 relations between the coupling constants in the maximal basis, and the minimal basis has only 435 couplings, the T-duality constraint (21) has no solution if one considers the effective action in this equation to be the minimal basis. In other words, the 468 relations imposed by T-duality are in general incompatible with the 435 couplings present in the minimal basis. This means that the effective action described by

the minimal basis cannot satisfy the T-duality constraint (21). The T-duality requirements introduce more constraints than there are free parameters in the minimal basis, resulting in an overconstrained system with no solution. To resolve this issue, one must work within a larger basis that has enough free parameters to accommodate the 468 T-duality relations. The maximal basis, which contains more couplings, provides the necessary degrees of freedom to find a consistent solution to the T-duality constraint.

The couplings in (27) are invariant under T-duality, with corresponding corrections at the 4-derivative order to the truncated Buscher rules. Since these corrections are very lengthy expressions, we do not write them here. We observed that for the case where all 333 parameters are zero, the correction $\Delta\alpha_{ij}^{(2)}$ has terms at the zeroth and the first orders of $\alpha^{ij}$.

# 4 T-duality on a basis with 468 couplings

In the previous section, we found that T-duality imposes 468 relations between the 801 couplings in the maximal basis. On the other hand, the minimal basis has 435 couplings, which is 33 less than the number of couplings required to be consistent with T-duality. Moreover, the T-duality constraints may fix some of the ambiguous coupling constants to be zero. Indeed, it is possible to find particular schemes where the minimal basis becomes fully consistent with T-duality. In such schemes, the T-duality constraints would fix the 33 ambiguous couplings absent in the minimal basis to vanish. However, finding such schemes is a nontrivial task.

As an alternative approach, we consider a basis that is neither the maximal nor the minimal one, but has 436 independent couplings. Starting from the original 2980 couplings in constructing the maximal basis, we add the field redefinition terms to them and remove the couplings with terms involving more than two derivatives, Ricci curvature, or scalar curvature. This constraint and some other constraints reduce the number of relations between the coupling constants to 468. Then, we choose a specific scheme among the remaining couplings, which can be described as follows:

$$
\begin{aligned}
\mathbf{S_1}^{(2)} = & -\frac{2\alpha'^2}{\kappa^2}\int d^{10}x\sqrt{-G}e^{-2\Phi}\Big[c_1 F_\alpha{}^{\gamma kl}F^{\alpha\beta ij}F_\beta{}^{\delta mn}F_\gamma{}^\epsilon{}_{mn}F_\delta{}^\varepsilon{}_{kl}F_{\epsilon\varepsilon ij} \\
& +c_2 F_\alpha{}^{\gamma kl}F^{\alpha\beta ij}F_\beta{}^\delta{}_k{}^m F_\gamma{}^\epsilon{}_m{}^n F_\delta{}^\varepsilon{}_{ln}F_{\epsilon\varepsilon ij} + c_3 F_\alpha{}^{\gamma kl}F^{\alpha\beta ij}F_\beta{}^\delta{}_{kl}F_\gamma{}^{\epsilon mn}F_\delta{}^\varepsilon{}_{mn}F_{\epsilon\varepsilon ij} + \cdots \\
& +c_{465}H_{\alpha\beta}{}^\delta H^{\alpha\beta\gamma}\nabla_\varepsilon H_{\delta\epsilon\mu}\nabla^\mu H_\gamma{}^{\epsilon\varepsilon} + c_{466}H_{\alpha\beta}{}^\delta H^{\alpha\beta\gamma}\nabla_\mu H_{\delta\epsilon\varepsilon}\nabla^\mu H_\gamma{}^{\epsilon\varepsilon} \\
& +c_{467}H_{\alpha\beta\gamma}H^{\alpha\beta\gamma}\nabla_\varepsilon H_{\delta\epsilon\mu}\nabla^\mu H^{\delta\epsilon\varepsilon} + c_{468}H_{\alpha\beta\gamma}H^{\alpha\beta\gamma}\nabla_\mu H_{\delta\epsilon\varepsilon}\nabla^\mu H^{\delta\epsilon\varepsilon}\Big] .
\end{aligned}
\tag{28}
$$

The expression above represents a subset of the 468 independent couplings, with the ellipsis symbolizing an additional 461 terms that are not explicitly listed.

The T-duality constraint (21) for the above couplings may not fix all the 468 couplings because some of the ambiguous couplings that T-duality fixes to zero may not be included in the above basis. However, the above basis must be consistent with the T-duality. We have found that the T-duality constraint (21) produces 416 relations between the 468 coupling constants and the fixed numbers of the lower-order action. This means the T-duality fixes 52

ambiguous coupling constants, which are not included in the above basis, to be zero. We have also observed that the remaining 52 coupling constants in the resulting T-duality invariant action can be removed by field redefinitions. So we are free to choose any values for these parameters. When all these parameters are set to zero, we find the following 107 non-zero couplings:

$$
\begin{aligned}
\mathbf{S_1}^{(2)} =\ & -\frac{2\alpha'^2}{8^2\kappa^2}\int d^{10}x\sqrt{-G}\,e^{-2\Phi}\Big[ -\frac{1}{8}F_{\alpha\beta ij}F^{\alpha\beta ij}F_\gamma{}^\epsilon{}_{kl}F^{\gamma\delta kl}F_\delta{}^{\varepsilon mn}F_{\epsilon\varepsilon mn} \\
& -2F_\alpha{}^\gamma{}_{ij}F^{\alpha\beta ij}F_\beta{}^{\delta kl}F^{\epsilon\varepsilon}{}_{kl}H_{\gamma\epsilon}{}^\mu H_{\delta\varepsilon\mu} +\frac{1}{8}F_{\alpha\beta ij}F^{\alpha\beta ij}F^{\gamma\delta kl}F^{\epsilon\varepsilon}{}_{kl}H_{\gamma\epsilon}{}^\mu H_{\delta\varepsilon\mu} \\
& +\frac{1}{2}F_\alpha{}^\gamma{}_{ij}F^{\alpha\beta ij}F_\beta{}^{\delta kl}F^{\epsilon\varepsilon}{}_{kl}H_{\gamma\delta}{}^\mu H_{\epsilon\varepsilon\mu} +\frac{1}{2}F_\alpha{}^\gamma{}_{ij}F^{\alpha\beta ij}F_\beta{}^{\delta kl}F_\gamma{}^\epsilon{}_{kl}H_\delta{}^{\varepsilon\mu}H_{\epsilon\varepsilon\mu} \\
& -F^{\alpha\beta ij}F^{\gamma\delta}{}_{ij}H_{\alpha\gamma}{}^\epsilon H_\beta{}^{\varepsilon\mu}H_{\delta\varepsilon}{}^\zeta H_{\epsilon\mu\zeta} +\frac{1}{2}F^{\alpha\beta ij}F^{\gamma\delta}{}_{ij}H_{\alpha\beta}{}^\epsilon H_\gamma{}^{\varepsilon\mu}H_{\delta\varepsilon}{}^\zeta H_{\epsilon\mu\zeta} \\
& +\frac{1}{48}F_{\alpha\beta ij}F^{\alpha\beta ij}H_\gamma{}^{\varepsilon\mu}H^{\gamma\delta\epsilon}H_{\delta\varepsilon}{}^\zeta H_{\epsilon\mu\zeta} +\frac{1}{24}F_\alpha{}^\gamma{}_{ij}F^{\alpha\beta ij}F_\beta{}^{\delta kl}F_{\gamma\delta kl}H_{\epsilon\varepsilon\mu}H^{\epsilon\varepsilon\mu} \\
& -\frac{1}{12}H_\alpha{}^{\delta\epsilon}H^{\alpha\beta\gamma}H_{\beta\delta}{}^\varepsilon H_\gamma{}^{\mu\zeta}H_{\epsilon\mu}{}^\eta H_{\varepsilon\zeta\eta} +\frac{1}{4}H_{\alpha\beta}{}^\delta H^{\alpha\beta\gamma}H_\gamma{}^{\epsilon\varepsilon}H_\delta{}^{\mu\zeta}H_{\epsilon\mu}{}^\eta H_{\varepsilon\zeta\eta} \\
& +\frac{19}{8}F_\alpha{}^\gamma{}_{ij}F^{\alpha\beta ij}H_\beta{}^{\delta\epsilon}H_{\gamma\delta}{}^\varepsilon H_\epsilon{}^{\mu\zeta}H_{\varepsilon\mu\zeta} +\frac{1}{32}F_{\alpha\beta ij}F^{\alpha\beta ij}H_{\gamma\delta}{}^\varepsilon H^{\gamma\delta\epsilon}H_\epsilon{}^{\mu\zeta}H_{\varepsilon\mu\zeta} \\
& +\frac{1}{48}F^{\alpha\beta ij}F^{\gamma\delta}{}_{ij}H_{\alpha\beta}{}^\epsilon H_{\gamma\delta\epsilon}H_{\varepsilon\mu\zeta}H^{\varepsilon\mu\zeta} +\frac{15}{16}H_{\alpha\beta}{}^\delta H^{\alpha\beta\gamma}H_\gamma{}^{\epsilon\varepsilon}H_{\delta\epsilon}{}^\mu H_\varepsilon{}^{\zeta\eta}H_{\mu\zeta\eta} \\
& -\frac{1}{96}H_{\alpha\beta\gamma}H^{\alpha\beta\gamma}H_{\delta\epsilon}{}^\mu H^{\delta\epsilon\varepsilon}H_\varepsilon{}^{\zeta\eta}H_{\mu\zeta\eta} -\frac{1}{24}F^{\alpha\beta ij}F^{\gamma\delta}{}_{ij}H_{\epsilon\varepsilon\mu}H^{\epsilon\varepsilon\mu}R_{\alpha\beta\gamma\delta} \\
& +2F^{\alpha\beta ij}F^{\gamma\delta}{}_{ij}H_\alpha{}^{\epsilon\varepsilon}H_{\gamma\epsilon}{}^\mu R_{\beta\delta\varepsilon\mu} -\frac{1}{8}F_{\alpha\beta ij}F^{\alpha\beta ij}F^{\gamma\delta kl}F^{\epsilon\varepsilon}{}_{kl}R_{\gamma\delta\epsilon\varepsilon} -4F_\alpha{}^\gamma{}_{ij}F^{\alpha\beta ij}R_\beta{}^{\delta\epsilon\varepsilon}R_{\gamma\delta\epsilon\varepsilon} \\
& -F^{\alpha\beta ij}F^{\gamma\delta}{}_{ij}H_\alpha{}^{\epsilon\varepsilon}H_{\beta\epsilon}{}^\mu R_{\gamma\delta\varepsilon\mu} +2F_\alpha{}^\gamma{}_{ij}F^{\alpha\beta ij}F_\beta{}^{\delta kl}F^{\epsilon\varepsilon}{}_{kl}R_{\gamma\epsilon\delta\varepsilon} -2F_\alpha{}^\gamma{}_{ij}F^{\alpha\beta ij}H_\beta{}^{\delta\epsilon}H_\delta{}^{\varepsilon\mu}R_{\gamma\epsilon\epsilon\mu} \\
& +2H_\alpha{}^{\delta\epsilon}H^{\alpha\beta\gamma}H_\beta{}^{\varepsilon\mu}H_{\delta\epsilon}{}^\zeta R_{\gamma\epsilon\mu\zeta} -2H_\alpha{}^{\delta\epsilon}H^{\alpha\beta\gamma}R_{\beta\delta}{}^{\varepsilon\mu}R_{\gamma\varepsilon\epsilon\mu} -8H_\alpha{}^{\delta\epsilon}H^{\alpha\beta\gamma}R_\beta{}^\varepsilon{}_\delta{}^\mu R_{\gamma\varepsilon\epsilon\mu} \\
& -\frac{1}{8}H_{\alpha\beta}{}^\delta H^{\alpha\beta\gamma}H_{\epsilon\varepsilon}{}^\zeta H^{\epsilon\varepsilon\mu}R_{\gamma\mu\delta\zeta} +\frac{1}{2}F_{\alpha\beta ij}F^{\alpha\beta ij}R_{\gamma\delta\epsilon\varepsilon}R^{\gamma\delta\epsilon\varepsilon} +2F^{\alpha\beta ij}F^{\gamma\delta}{}_{ij}H_{\alpha\gamma}{}^\epsilon H_\beta{}^{\varepsilon\mu}R_{\delta\epsilon\varepsilon\mu} \\
& +F_\alpha{}^\gamma{}_{ij}F^{\alpha\beta ij}H_\beta{}^{\delta\epsilon}H_\gamma{}^{\varepsilon\mu}R_{\delta\epsilon\varepsilon\mu} -\frac{5}{8}F_{\alpha\beta ij}F^{\alpha\beta ij}H_\gamma{}^{\varepsilon\mu}H^{\gamma\delta\epsilon}R_{\delta\epsilon\varepsilon\mu} -2F^{\alpha\beta ij}F^{\gamma\delta}{}_{ij}H_{\alpha\beta}{}^\epsilon H_\gamma{}^{\varepsilon\mu}R_{\delta\varepsilon\epsilon\mu} \\
& +6H_\alpha{}^{\delta\epsilon}H^{\alpha\beta\gamma}R_\beta{}^\varepsilon{}_\gamma{}^\mu R_{\delta\varepsilon\epsilon\mu} -\frac{13}{2}H_\alpha{}^{\delta\epsilon}H^{\alpha\beta\gamma}H_{\beta\delta}{}^\varepsilon H_\gamma{}^{\mu\zeta}R_{\epsilon\varepsilon\mu\zeta} -\frac{11}{8}H_{\alpha\beta}{}^\delta H^{\alpha\beta\gamma}H_\gamma{}^{\epsilon\varepsilon}H_\delta{}^{\mu\zeta}R_{\epsilon\varepsilon\mu\zeta} \\
& +\frac{1}{24}H_{\alpha\beta\gamma}H^{\alpha\beta\gamma}H_\delta{}^{\mu\zeta}H^{\delta\epsilon\varepsilon}R_{\epsilon\varepsilon\mu\zeta} -\frac{1}{12}F^{\alpha\beta ij}H_{\beta\gamma\delta}H_{\epsilon\varepsilon\mu}H^{\epsilon\varepsilon\mu}\nabla_\alpha F^{\gamma\delta}{}_{ij} \\
& +6F^{\alpha\beta ij}H_\gamma{}^{\epsilon\varepsilon}R_{\beta\delta\epsilon\varepsilon}\nabla_\alpha F^{\gamma\delta}{}_{ij} +3F^{\alpha\beta ij}H_\beta{}^{\epsilon\varepsilon}R_{\gamma\delta\epsilon\varepsilon}\nabla_\alpha F^{\gamma\delta}{}_{ij} -4F_\alpha{}^{\gamma kl}F^{\alpha\beta ij}F^{\delta\epsilon}{}_{ij}H_{\gamma\epsilon\varepsilon}\nabla_\beta F_\delta{}^\varepsilon{}_{kl} \\
& -2F_\alpha{}^\gamma{}_{ij}F^{\alpha\beta ij}F^{\delta\epsilon kl}H_{\gamma\epsilon\varepsilon}\nabla_\beta F_\delta{}^\varepsilon{}_{kl} -F^{\alpha\beta ij}F^{\gamma\delta}{}_{ij}F^{\epsilon\varepsilon kl}H_{\beta\delta\varepsilon}\nabla_\gamma F_{\alpha\epsilon kl} \\
& -2F^{\alpha\beta ij}F^{\gamma\delta kl}\nabla_\beta F_{\delta\epsilon kl}\nabla_\gamma F_\alpha{}^\epsilon{}_{ij} +2F_\alpha{}^{\gamma kl}F^{\alpha\beta ij}\nabla_\beta F^{\delta\epsilon}{}_{ij}\nabla_\gamma F_{\delta\epsilon kl} \\
& -2F_\alpha{}^\gamma{}_{ij}F^{\alpha\beta ij}\nabla_\beta F^{\delta\epsilon kl}\nabla_\gamma F_{\delta\epsilon kl} +F_\alpha{}^{\gamma kl}F^{\alpha\beta ij}\nabla_\beta F_{\delta\epsilon kl}\nabla_\gamma F^{\delta\epsilon}{}_{ij} \\
& -2F_\alpha{}^{\gamma kl}F^{\alpha\beta ij}F_\beta{}^\delta{}_{kl}H_{\delta\epsilon\varepsilon}\nabla_\gamma F^{\epsilon\varepsilon}{}_{ij} +2F_\alpha{}^\gamma{}_{ij}F^{\alpha\beta ij}F_\beta{}^{\delta kl}H_{\delta\epsilon\varepsilon}\nabla_\gamma F^{\epsilon\varepsilon}{}_{kl}
\end{aligned}
$$

$$+F_{\alpha\beta}{}^{kl}F^{\alpha\beta ij}F^{\gamma\delta}{}_{ij}H_{\delta\epsilon\varepsilon}\nabla_\gamma F^{\epsilon\varepsilon}{}_{kl}-\frac{1}{4}F_{\alpha\beta ij}F^{\alpha\beta ij}F^{\gamma\delta kl}H_{\delta\epsilon\varepsilon}\nabla_\gamma F^{\epsilon\varepsilon}{}_{kl} \tag{29}$$

$$-2H_\alpha{}^{\delta\epsilon}H^{\alpha\beta\gamma}R_{\delta\epsilon\varepsilon\mu}\nabla_\gamma H_\beta{}^{\varepsilon\mu}-\frac{1}{2}F_\alpha{}^\gamma{}_{ij}F^{\alpha\beta ij}F_\beta{}^{\delta kl}F^{\epsilon\varepsilon}{}_{kl}\nabla_\gamma H_{\delta\epsilon\varepsilon}$$

$$-\frac{2}{3}F_\alpha{}^\gamma{}_{ij}F^{\alpha\beta ij}\nabla_\beta H^{\delta\epsilon\varepsilon}\nabla_\gamma H_{\delta\epsilon\varepsilon}+\frac{1}{12}H_{\delta\epsilon\varepsilon}H^{\delta\epsilon\varepsilon}\nabla_\beta F_{\alpha\gamma ij}\nabla^\gamma F^{\alpha\beta ij}$$

$$+4F_\alpha{}^{\gamma kl}F^{\alpha\beta ij}F^{\delta\epsilon}{}_{ij}H_{\gamma\epsilon\varepsilon}\nabla_\delta F_\beta{}^\varepsilon{}_{kl}+F_\alpha{}^\gamma{}_{ij}F^{\alpha\beta ij}F^{\delta\epsilon kl}H_{\gamma\epsilon\varepsilon}\nabla_\delta F_\beta{}^\varepsilon{}_{kl}$$

$$-2F^{\alpha\beta ij}F^{\gamma\delta kl}\nabla_\beta F_\alpha{}^\epsilon{}_{kl}\nabla_\delta F_{\gamma\epsilon ij}-3F^{\alpha\beta ij}F^{\gamma\delta kl}\nabla_\beta F_\alpha{}^\epsilon{}_{ij}\nabla_\delta F_{\gamma\epsilon kl}$$

$$-F_\alpha{}^{\gamma kl}F^{\alpha\beta ij}F^{\delta\epsilon}{}_{ij}H_{\beta\epsilon\varepsilon}\nabla_\delta F_\gamma{}^\varepsilon{}_{kl}-2H^{\alpha\beta\gamma}H^{\delta\epsilon\varepsilon}R_{\gamma\epsilon\varepsilon\mu}\nabla_\delta H_{\alpha\beta}{}^\mu+\frac{1}{4}H^{\alpha\beta\gamma}H^{\delta\epsilon\varepsilon}\nabla_\gamma H_{\epsilon\varepsilon\mu}\nabla_\delta H_{\alpha\beta}{}^\mu$$

$$+\frac{9}{4}F^{\alpha\beta ij}F^{\gamma\delta}{}_{ij}H_{\alpha\gamma}{}^\epsilon H_\epsilon{}^{\varepsilon\mu}\nabla_\delta H_{\beta\varepsilon\mu}+H_\alpha{}^{\delta\epsilon}H^{\alpha\beta\gamma}\nabla_\gamma H_{\epsilon\varepsilon\mu}\nabla_\delta H_\beta{}^{\varepsilon\mu}$$

$$+\frac{3}{2}F_\alpha{}^\gamma{}_{ij}F^{\alpha\beta ij}F_\beta{}^{\delta kl}F^{\epsilon\varepsilon}{}_{kl}\nabla_\delta H_{\gamma\epsilon\varepsilon}-\frac{9}{8}F^{\alpha\beta ij}F^{\gamma\delta}{}_{ij}H_{\alpha\beta}{}^\epsilon H_\epsilon{}^{\varepsilon\mu}\nabla_\delta H_{\gamma\varepsilon\mu}$$

$$+\frac{7}{4}F^{\alpha\beta ij}F^{\gamma\delta}{}_{ij}H_{\alpha\gamma}{}^\epsilon H_\beta{}^{\varepsilon\mu}\nabla_\delta H_{\epsilon\varepsilon\mu}-\frac{7}{8}F^{\alpha\beta ij}F^{\gamma\delta}{}_{ij}H_{\alpha\beta}{}^\epsilon H_\gamma{}^{\varepsilon\mu}\nabla_\delta H_{\epsilon\varepsilon\mu}$$

$$+\frac{1}{12}F^{\alpha\beta ij}F^{\gamma\delta}{}_{ij}H_{\alpha\beta\gamma}H^{\epsilon\varepsilon\mu}\nabla_\delta H_{\epsilon\varepsilon\mu}-6F^{\alpha\beta ij}H_\delta{}^{\epsilon\varepsilon}R_{\beta\gamma\epsilon\varepsilon}\nabla^\delta F_\alpha{}^\gamma{}_{ij}+8R_{\alpha\delta}{}^{\epsilon\varepsilon}R_{\beta\epsilon\gamma\varepsilon}\nabla^\delta H^{\alpha\beta\gamma}$$

$$+3R_{\beta\gamma\epsilon\varepsilon}\nabla_\alpha H_\delta{}^{\epsilon\varepsilon}\nabla^\delta H^{\alpha\beta\gamma}-R_{\beta\gamma\epsilon\varepsilon}\nabla_\delta H_\alpha{}^{\epsilon\varepsilon}\nabla^\delta H^{\alpha\beta\gamma}-4F_\alpha{}^{\gamma kl}F^{\alpha\beta ij}F^{\delta\epsilon}{}_{ij}H_{\beta\gamma\varepsilon}\nabla_\epsilon F_\delta{}^\varepsilon{}_{kl}$$

$$-\frac{13}{2}H_\alpha{}^{\delta\epsilon}H^{\alpha\beta\gamma}\nabla_\delta H_\beta{}^{\varepsilon\mu}\nabla_\epsilon H_{\gamma\varepsilon\mu}-2F^{\alpha\beta ij}F^{\gamma\delta kl}\nabla_\beta F_{\delta\epsilon kl}\nabla^\epsilon F_{\alpha\gamma ij}$$

$$+4F^{\alpha\beta ij}F^{\gamma\delta kl}\nabla_\delta F_{\beta\epsilon kl}\nabla^\epsilon F_{\alpha\gamma ij}+2F_\alpha{}^{\gamma kl}F^{\alpha\beta ij}\nabla_\delta F_{\gamma\epsilon ij}\nabla^\epsilon F_\beta{}^\delta{}_{kl}$$

$$+2F_\alpha{}^\gamma{}_{ij}F^{\alpha\beta ij}\nabla_\delta F_{\gamma\epsilon kl}\nabla^\epsilon F_\beta{}^{\delta kl}+\frac{1}{2}F^{\alpha\beta ij}H_{\alpha\gamma\epsilon}H_\beta{}^{\varepsilon\mu}H_{\delta\varepsilon\mu}\nabla^\epsilon F^{\gamma\delta}{}_{ij}$$

$$+\frac{1}{4}F^{\alpha\beta ij}H_{\alpha\beta\gamma}H_\delta{}^{\varepsilon\mu}H_{\epsilon\varepsilon\mu}\nabla^\epsilon F^{\gamma\delta}{}_{ij}+4F^{\alpha\beta ij}H_{\gamma\epsilon}{}^\varepsilon R_{\alpha\beta\delta\varepsilon}\nabla^\epsilon F^{\gamma\delta}{}_{ij}$$

$$-8F^{\alpha\beta ij}H_{\alpha\gamma}{}^\varepsilon R_{\beta\delta\epsilon\varepsilon}\nabla^\epsilon F^{\gamma\delta}{}_{ij}+2F^{\alpha\beta ij}H_{\alpha\beta}{}^\varepsilon R_{\gamma\delta\epsilon\varepsilon}\nabla^\epsilon F^{\gamma\delta}{}_{ij}-F_{\alpha\beta}{}^{kl}F^{\alpha\beta ij}\nabla_\delta F_{\gamma\epsilon kl}\nabla^\epsilon F^{\gamma\delta}{}_{ij}$$

$$+\frac{1}{4}F_{\alpha\beta ij}F^{\alpha\beta ij}\nabla_\delta F_{\gamma\epsilon kl}\nabla^\epsilon F^{\gamma\delta kl}-3F^{\alpha\beta ij}F^{\gamma\delta}{}_{ij}F^{\epsilon\varepsilon kl}H_{\beta\gamma\delta}\nabla_\varepsilon F_{\alpha\epsilon kl}$$

$$-2H^{\alpha\beta\gamma}H^{\delta\epsilon\varepsilon}\nabla_\beta H_{\alpha\delta}{}^\mu\nabla_\varepsilon H_{\gamma\epsilon\mu}+\frac{15}{2}H^{\alpha\beta\gamma}H^{\delta\epsilon\varepsilon}\nabla_\delta H_{\alpha\beta}{}^\mu\nabla_\varepsilon H_{\gamma\epsilon\mu}$$

$$-\frac{3}{8}H^{\alpha\beta\gamma}H^{\delta\epsilon\varepsilon}\nabla_\gamma H_{\alpha\beta}{}^\mu\nabla_\varepsilon H_{\delta\epsilon\mu}+\frac{1}{2}H_{\alpha\beta}{}^\delta H^{\alpha\beta\gamma}H_\gamma{}^{\epsilon\varepsilon}H_\epsilon{}^{\mu\zeta}\nabla_\varepsilon H_{\delta\mu\zeta}$$

$$-\frac{1}{2}H_\alpha{}^{\delta\epsilon}H^{\alpha\beta\gamma}H_{\beta\delta}{}^\varepsilon H_\gamma{}^{\mu\zeta}\nabla_\varepsilon H_{\epsilon\mu\zeta}-2R_{\gamma\delta\epsilon\varepsilon}\nabla^\delta H^{\alpha\beta\gamma}\nabla^\varepsilon H_{\alpha\beta}{}^\epsilon+2\nabla^\delta H^{\alpha\beta\gamma}\nabla_\epsilon H_{\gamma\delta\varepsilon}\nabla^\varepsilon H_{\alpha\beta}{}^\epsilon$$

$$-4\nabla^\delta H^{\alpha\beta\gamma}\nabla_\varepsilon H_{\gamma\delta\epsilon}\nabla^\varepsilon H_{\alpha\beta}{}^\epsilon+4R_{\beta\gamma\epsilon\varepsilon}\nabla^\delta H^{\alpha\beta\gamma}\nabla^\varepsilon H_{\alpha\delta}{}^\epsilon-4\nabla_\gamma H_{\beta\epsilon\varepsilon}\nabla^\delta H^{\alpha\beta\gamma}\nabla^\varepsilon H_{\alpha\delta}{}^\epsilon$$

$$+4F_\alpha{}^\gamma{}_{ij}F^{\alpha\beta ij}R_{\gamma\delta\epsilon\varepsilon}\nabla^\varepsilon H_\beta{}^{\delta\epsilon}+2F_\alpha{}^\gamma{}_{ij}F^{\alpha\beta ij}\nabla_\epsilon H_{\gamma\delta\varepsilon}\nabla^\varepsilon H_\beta{}^{\delta\epsilon}+\frac{3}{8}F_{\alpha\beta ij}F^{\alpha\beta ij}\nabla_\epsilon H_{\gamma\delta\varepsilon}\nabla^\varepsilon H^{\gamma\delta\epsilon}$$

$$-\frac{7}{2}H_\alpha{}^{\delta\epsilon}H^{\alpha\beta\gamma}\nabla_\epsilon H_{\delta\varepsilon\mu}\nabla^\mu H_{\beta\gamma}{}^\varepsilon+H_\alpha{}^{\delta\epsilon}H^{\alpha\beta\gamma}\nabla_\mu H_{\gamma\epsilon\varepsilon}\nabla^\mu H_{\beta\delta}{}^\varepsilon-2H_{\alpha\beta}{}^\delta H^{\alpha\beta\gamma}R_{\delta\epsilon\varepsilon\mu}\nabla^\mu H_\gamma{}^{\epsilon\varepsilon}$$

$$+\frac{1}{8}H_{\alpha\beta}{}^\delta H^{\alpha\beta\gamma}\nabla_\delta H_{\epsilon\varepsilon\mu}\nabla^\mu H_\gamma{}^{\epsilon\varepsilon}+\frac{17}{4}H_{\alpha\beta}{}^\delta H^{\alpha\beta\gamma}\nabla_\varepsilon H_{\delta\epsilon\mu}\nabla^\mu H_\gamma{}^{\epsilon\varepsilon}$$

$$-\frac{1}{4}H_{\alpha\beta}{}^{\delta}H^{\alpha\beta\gamma}\nabla_{\mu}H_{\delta\epsilon\varepsilon}\nabla^{\mu}H_{\gamma}{}^{\epsilon\varepsilon}-\frac{1}{8}H_{\alpha\beta\gamma}H^{\alpha\beta\gamma}\nabla_{\varepsilon}H_{\delta\epsilon\mu}\nabla^{\mu}H^{\delta\epsilon\varepsilon}$$

$$+\frac{1}{36}H_{\alpha\beta\gamma}H^{\alpha\beta\gamma}\nabla_{\mu}H_{\delta\epsilon\varepsilon}\nabla^{\mu}H^{\delta\epsilon\varepsilon}\bigg].$$

Note that there are three-field couplings in the above action. We have checked that the above couplings and the couplings in (27) are the same up to appropriate field redefinitions. The above couplings are manifestly invariant under T-duality, with some lengthy expressions for the 4-derivative corrections to the truncated Buscher rules. For example, the correction term $\Delta\alpha_{ij}^{(2)}$ has 292 non-zero terms, which include the zeroth and first order contributions in the scalar $\alpha^{ij}$. We do not write these expressions explicitly, as they are quite lengthy. These 4-derivative corrections are needed if one would like to find 8-derivative couplings by applying T-duality, which is not the focus of our current interest.

# 5    Couplings in Canonical form

Having found the T-duality invariant couplings with fixed coupling constants in (27) or in (29), one may use field redefinition to rewrite them in a canonical form where the dilaton appears only as the overall factor $e^{-2\Phi}$, and the couplings have no Ricci or scalar curvature, no first derivative of Riemann curvature, no second derivative of the $H$-field and $F$-field, and no three-field couplings.

To find the couplings in this form, we add the total derivative terms, field redefinition terms, and the terms from the $H$-field Bianchi identities to the most general coupling, which has 2980 couplings. We then equate them with the 260 couplings in (27) or 107 couplings in (29) that the T-duality produces. We go to the local frames in both external and internal spaces to impose the remaining Bianchi identities.

If one sets to zero some of the 2980 couplings in the resulting equation and the equation has a solution, then that choice is allowed. In this way, we can write the couplings in the canonical form. Imposing the canonical form for the 2980 couplings, one finds the equation has a solution, and there are still some unfixed parameters. We choose them to write the couplings in the following 85 couplings:

$$\mathbf{S_1}^{(2)} = -\frac{2\alpha'^2}{8^2\kappa^2}\int d^{10}x\sqrt{-G}e^{-2\Phi}\Big[[\text{NS--NS}]_{10} + [F^4H^2]_{12} + [F^2H^4]_6 + [F^2H^2R]_9 + [F^3H\nabla F]_{15}$$

$$+[F^4R]_4 + [F^2H^2\nabla H]_8 + [F^2R\nabla H]_3 + [F^2(\nabla F)^2]_9 + [F^2(\nabla H)^2]_5 + [F^2R^2]_4\Big]. \quad (30)$$

The NS-NS couplings mentioned are those that have been found in [11] by studying the T-duality of only NS-NS fields, and written in canonical form in [23]. We have also used an identity to write the 11 terms reported in [23] in terms of the following 10 terms:

$$[\text{NS--NS}]_{10} = 2H^{\alpha\beta\gamma}H^{\delta\epsilon\varepsilon}R_{\alpha\beta\delta}{}^{\mu}R_{\gamma\mu\epsilon\varepsilon} - \frac{1}{12}H_{\alpha}{}^{\delta\epsilon}H^{\alpha\beta\gamma}H_{\beta\delta}{}^{\varepsilon}H_{\gamma}{}^{\mu\zeta}H_{\epsilon\mu}{}^{\eta}H_{\varepsilon\zeta\eta} - 2H_{\alpha}{}^{\delta\epsilon}H^{\alpha\beta\gamma}R_{\beta\delta}{}^{\varepsilon\mu}R_{\gamma\varepsilon\epsilon\mu}$$

$$-2H_{\alpha}{}^{\delta\epsilon}H^{\alpha\beta\gamma}R_{\beta}{}^{\varepsilon}{}_{\gamma}{}^{\mu}R_{\delta\varepsilon\epsilon\mu} + H_{\alpha}{}^{\delta\epsilon}H^{\alpha\beta\gamma}H_{\beta\delta}{}^{\varepsilon}H_{\gamma}{}^{\mu\zeta}R_{\epsilon\varepsilon\mu\zeta} - 4H^{\alpha\beta\gamma}H^{\delta\epsilon\varepsilon}R_{\gamma\epsilon\varepsilon\mu}\nabla_{\beta}H_{\alpha\delta}{}^{\mu}$$

$$-H_\alpha{}^{\delta\epsilon}H^{\alpha\beta\gamma}R_{\delta\epsilon\varepsilon\mu}\nabla_\gamma H_\beta{}^{\varepsilon\mu} - \frac{1}{2}H^{\alpha\beta\gamma}H^{\delta\epsilon\varepsilon}\nabla_\beta H_{\alpha\delta}{}^\mu\nabla_\varepsilon H_{\gamma\epsilon\mu}$$

$$-\frac{1}{2}H_\alpha{}^{\delta\epsilon}H^{\alpha\beta\gamma}H_{\beta\delta}{}^\varepsilon H_\gamma{}^{\mu\zeta}\nabla_\varepsilon H_{\epsilon\mu\zeta} + \frac{1}{4}H_\alpha{}^{\delta\epsilon}H^{\alpha\beta\gamma}\nabla_\epsilon H_{\delta\varepsilon\mu}\nabla^\mu H_{\beta\gamma}{}^\varepsilon \,. \tag{31}$$

By expressing the 11 terms from the previous work in this more compact set of 10 terms, we have simplified the representation of the NS-NS couplings in the canonical form. The other couplings that involve YM fields are:

$$
\begin{aligned}
[F^4H^2]_{12} &= \frac{1}{8}F_\alpha{}^{\gamma kl}F^{\alpha\beta ij}F^{\delta\epsilon}{}_{ij}F^{\varepsilon\mu}{}_{kl}H_{\beta\epsilon\mu}H_{\gamma\delta\epsilon} + \frac{3}{2}F_\alpha{}^{\gamma kl}F^{\alpha\beta ij}F^{\delta\epsilon}{}_{ij}F^{\varepsilon\mu}{}_{kl}H_{\beta\delta\epsilon}H_{\gamma\epsilon\mu} \\
&\quad -\frac{3}{2}F_\alpha{}^\gamma{}_{ij}F^{\alpha\beta ij}F^{\delta\epsilon kl}F^{\varepsilon\mu}{}_{kl}H_{\beta\delta\varepsilon}H_{\gamma\epsilon\mu} - \frac{1}{2}F_\alpha{}^{\gamma kl}F^{\alpha\beta ij}F^{\delta\epsilon}{}_{ij}F^{\varepsilon\mu}{}_{kl}H_{\beta\delta\epsilon}H_{\gamma\varepsilon\mu} \\
&\quad -\frac{1}{4}F_\alpha{}^\gamma{}_{ij}F^{\alpha\beta ij}F^{\delta\epsilon kl}F^{\varepsilon\mu}{}_{kl}H_{\beta\delta\epsilon}H_{\gamma\varepsilon\mu} - F_\alpha{}^{\gamma kl}F^{\alpha\beta ij}F_\delta{}^\varepsilon{}_{kl}F^{\delta\epsilon}{}_{ij}H_{\beta\epsilon}{}^\mu H_{\gamma\varepsilon\mu} \\
&\quad -2F_\alpha{}^\gamma{}_{ij}F^{\alpha\beta ij}F_\delta{}^\varepsilon{}_{kl}F^{\delta\epsilon kl}H_{\beta\epsilon}{}^\mu H_{\gamma\varepsilon\mu} - 2F_\alpha{}^\gamma{}_{ij}F^{\alpha\beta ij}F_\beta{}^{\delta kl}F^{\epsilon\varepsilon}{}_{kl}H_{\gamma\epsilon}{}^\mu H_{\delta\varepsilon\mu} \\
&\quad +\frac{1}{8}F_{\alpha\beta ij}F^{\alpha\beta ij}F^{\gamma\delta kl}F^{\epsilon\varepsilon}{}_{kl}H_{\gamma\epsilon}{}^\mu H_{\delta\varepsilon\mu} - \frac{1}{16}F_{\alpha\beta ij}F^{\alpha\beta ij}F^{\gamma\delta kl}F^{\epsilon\varepsilon}{}_{kl}H_{\gamma\delta}{}^\mu H_{\epsilon\varepsilon\mu} \\
&\quad +\frac{1}{2}F_\alpha{}^\gamma{}_{ij}F^{\alpha\beta ij}F_\beta{}^{\delta kl}F_\gamma{}^\epsilon{}_{kl}H_\delta{}^{\varepsilon\mu}H_{\epsilon\varepsilon\mu} + \frac{1}{8}F_{\alpha\beta ij}F^{\alpha\beta ij}F_\gamma{}^\epsilon{}_{kl}F^{\gamma\delta kl}H_\delta{}^{\varepsilon\mu}H_{\epsilon\varepsilon\mu}\,, \\[6pt]
[F^2H^4]_6 &= -F^{\alpha\beta ij}F^{\gamma\delta}{}_{ij}H_{\alpha\gamma}{}^\epsilon H_\beta{}^{\varepsilon\mu}H_{\delta\varepsilon}{}^\zeta H_{\epsilon\mu\zeta} + \frac{1}{2}F^{\alpha\beta ij}F^{\gamma\delta}{}_{ij}H_{\alpha\beta}{}^\epsilon H_\gamma{}^{\varepsilon\mu}H_{\delta\varepsilon}{}^\zeta H_{\epsilon\mu\zeta} \\
&\quad -\frac{1}{2}F_\alpha{}^\gamma{}_{ij}F^{\alpha\beta ij}H_\beta{}^{\delta\epsilon}H_\gamma{}^{\varepsilon\mu}H_{\delta\varepsilon}{}^\zeta H_{\epsilon\mu\zeta} + \frac{1}{48}F_{\alpha\beta ij}F^{\alpha\beta ij}H_\gamma{}^{\varepsilon\mu}H^{\gamma\delta\epsilon}H_{\delta\varepsilon}{}^\zeta H_{\epsilon\mu\zeta} \\
&\quad -\frac{1}{2}F_\alpha{}^\gamma{}_{ij}F^{\alpha\beta ij}H_\beta{}^{\delta\epsilon}H_{\gamma\delta}{}^\varepsilon H_\epsilon{}^{\mu\zeta}H_{\varepsilon\mu\zeta} + \frac{1}{16}F_{\alpha\beta ij}F^{\alpha\beta ij}H_{\gamma\delta}{}^\varepsilon H^{\gamma\delta\epsilon}H_\epsilon{}^{\mu\zeta}H_{\varepsilon\mu\zeta}\,, \\[6pt]
[F^2H^2R]_9 &= 2F^{\alpha\beta ij}F^{\gamma\delta}{}_{ij}H_\alpha{}^{\epsilon\varepsilon}H_{\gamma\epsilon}{}^\mu R_{\beta\delta\varepsilon\mu} + F^{\alpha\beta ij}F^{\gamma\delta}{}_{ij}H_{\alpha\gamma}{}^\epsilon H_\epsilon{}^{\varepsilon\mu}R_{\beta\delta\varepsilon\mu} \\
&\quad -F^{\alpha\beta ij}F^{\gamma\delta}{}_{ij}H_\alpha{}^{\epsilon\varepsilon}H_{\beta\epsilon}{}^\mu R_{\gamma\delta\varepsilon\mu} - \frac{1}{2}F^{\alpha\beta ij}F^{\gamma\delta}{}_{ij}H_{\alpha\beta}{}^\epsilon H_\epsilon{}^{\varepsilon\mu}R_{\gamma\delta\varepsilon\mu} \\
&\quad -6F_\alpha{}^\gamma{}_{ij}F^{\alpha\beta ij}H_\beta{}^{\delta\epsilon}H_\delta{}^{\varepsilon\mu}R_{\gamma\epsilon\varepsilon\mu} + 2F^{\alpha\beta ij}F^{\gamma\delta}{}_{ij}H_{\alpha\gamma}{}^\epsilon H_\beta{}^{\varepsilon\mu}R_{\delta\epsilon\varepsilon\mu} \\
&\quad +2F_\alpha{}^\gamma{}_{ij}F^{\alpha\beta ij}H_\beta{}^{\delta\epsilon}H_\gamma{}^{\varepsilon\mu}R_{\delta\epsilon\varepsilon\mu} - \frac{3}{4}F_{\alpha\beta ij}F^{\alpha\beta ij}H_\gamma{}^{\varepsilon\mu}H^{\gamma\delta\epsilon}R_{\delta\epsilon\varepsilon\mu} \\
&\quad -2F^{\alpha\beta ij}F^{\gamma\delta}{}_{ij}H_{\alpha\beta}{}^\epsilon H_\gamma{}^{\varepsilon\mu}R_{\delta\varepsilon\epsilon\mu}\,, \\[6pt]
[F^4R]_4 &= 2F_\alpha{}^{\gamma kl}F^{\alpha\beta ij}F_\delta{}^\varepsilon{}_{kl}F^{\delta\epsilon}{}_{ij}R_{\beta\epsilon\gamma\varepsilon} + 2F_\alpha{}^\gamma{}_{ij}F^{\alpha\beta ij}F_\delta{}^\varepsilon{}_{kl}F^{\delta\epsilon kl}R_{\beta\epsilon\gamma\varepsilon} \\
&\quad -\frac{1}{4}F_{\alpha\beta ij}F^{\alpha\beta ij}F^{\gamma\delta kl}F^{\epsilon\varepsilon}{}_{kl}R_{\gamma\delta\epsilon\varepsilon} - 2F_\alpha{}^{\gamma kl}F^{\alpha\beta ij}F_\beta{}^\delta{}_{kl}F^{\epsilon\varepsilon}{}_{ij}R_{\gamma\epsilon\delta\varepsilon}\,, \\[6pt]
[F^2H^2\nabla H]_8 &= -2F^{\alpha\beta ij}F^{\gamma\delta}{}_{ij}H_\alpha{}^{\epsilon\varepsilon}H_{\gamma\epsilon}{}^\mu\nabla_\delta H_{\beta\varepsilon\mu} - \frac{3}{4}F^{\alpha\beta ij}F^{\gamma\delta}{}_{ij}H_{\alpha\gamma}{}^\epsilon H_\epsilon{}^{\varepsilon\mu}\nabla_\delta H_{\beta\varepsilon\mu} \\
&\quad +F^{\alpha\beta ij}F^{\gamma\delta}{}_{ij}H_\alpha{}^{\epsilon\varepsilon}H_{\beta\epsilon}{}^\mu\nabla_\delta H_{\gamma\varepsilon\mu} + \frac{3}{8}F^{\alpha\beta ij}F^{\gamma\delta}{}_{ij}H_{\alpha\beta}{}^\epsilon H_\epsilon{}^{\varepsilon\mu}\nabla_\delta H_{\gamma\varepsilon\mu} \\
&\quad +\frac{1}{2}F^{\alpha\beta ij}F^{\gamma\delta}{}_{ij}H_{\alpha\gamma}{}^\epsilon H_\beta{}^{\varepsilon\mu}\nabla_\delta H_{\epsilon\varepsilon\mu} + 2F_\alpha{}^\gamma{}_{ij}F^{\alpha\beta ij}H_\beta{}^{\delta\epsilon}H_\delta{}^{\varepsilon\mu}\nabla_\epsilon H_{\gamma\varepsilon\mu} \\
&\quad -\frac{1}{2}F^{\alpha\beta ij}F^{\gamma\delta}{}_{ij}H_{\alpha\gamma}{}^\epsilon H_\beta{}^{\varepsilon\mu}\nabla_\epsilon H_{\delta\varepsilon\mu} - \frac{1}{2}F^{\alpha\beta ij}F^{\gamma\delta}{}_{ij}H_{\alpha\beta}{}^\epsilon H_\gamma{}^{\varepsilon\mu}\nabla_\mu H_{\delta\epsilon\varepsilon}\,,
\end{aligned}
$$

$$
\begin{aligned}
[F^3 H\nabla F]_{15} &= -F^{\alpha\beta ij}F^{\gamma\delta}{}_{ij}F^{\epsilon\varepsilon kl}H_{\delta\epsilon}\nabla_\beta F_{\alpha\gamma kl} - F^{\alpha\beta ij}F^{\gamma\delta}{}_{ij}F^{\epsilon\varepsilon kl}H_{\gamma\delta\varepsilon}\nabla_\beta F_{\alpha\epsilon kl} \\
&\quad -2F_\alpha{}^{\gamma kl}F^{\alpha\beta ij}F^{\delta\epsilon}{}_{ij}H_{\gamma\epsilon\varepsilon}\nabla_\beta F_\delta{}^\varepsilon{}_{kl} + 4F_\alpha{}^\gamma{}_{ij}F^{\alpha\beta ij}F^{\delta\epsilon kl}H_{\gamma\epsilon\varepsilon}\nabla_\beta F_\delta{}^\varepsilon{}_{kl} \\
&\quad +6F^{\alpha\beta ij}F^{\gamma\delta}{}_{ij}F^{\epsilon\varepsilon kl}H_{\beta\delta\varepsilon}\nabla_\gamma F_{\alpha\epsilon kl} + 2F_\alpha{}^\gamma{}_{ij}F^{\alpha\beta ij}F^{\delta\epsilon kl}H_{\delta\epsilon\varepsilon}\nabla_\gamma F_\beta{}^\varepsilon{}_{kl} \\
&\quad -2F_\alpha{}^{\gamma kl}F^{\alpha\beta ij}F^{\delta\epsilon}{}_{ij}H_{\beta\epsilon\varepsilon}\nabla_\gamma F_\delta{}^\varepsilon{}_{kl} - F_\alpha{}^{\gamma kl}F^{\alpha\beta ij}F_\beta{}^\delta{}_{kl}H_{\delta\epsilon\varepsilon}\nabla_\gamma F^{\epsilon\varepsilon}{}_{ij} \\
&\quad +3F_\alpha{}^\gamma{}_{ij}F^{\alpha\beta ij}F_\beta{}^{\delta kl}H_{\delta\epsilon\varepsilon}\nabla_\gamma F^{\epsilon\varepsilon}{}_{kl} - \frac{1}{4}F_{\alpha\beta ij}F^{\alpha\beta ij}F^{\gamma\delta kl}H_{\delta\epsilon\varepsilon}\nabla_\gamma F^{\epsilon\varepsilon}{}_{kl} \\
&\quad -6F_\alpha{}^\gamma{}_{ij}F^{\alpha\beta ij}F^{\delta\epsilon kl}H_{\gamma\epsilon\varepsilon}\nabla_\delta F_\beta{}^\varepsilon{}_{kl} + 2F_\alpha{}^{\gamma kl}F^{\alpha\beta ij}F^{\delta\epsilon}{}_{ij}H_{\beta\epsilon\varepsilon}\nabla_\delta F_\gamma{}^\varepsilon{}_{kl} \\
&\quad +F_\alpha{}^\gamma{}_{ij}F^{\alpha\beta ij}F_\beta{}^{\delta kl}H_{\gamma\epsilon\varepsilon}\nabla_\delta F^{\epsilon\varepsilon}{}_{kl} - 2F_\alpha{}^{\gamma kl}F^{\alpha\beta ij}F^{\delta\epsilon}{}_{ij}H_{\beta\gamma\varepsilon}\nabla_\epsilon F_\delta{}^\varepsilon{}_{kl} \\
&\quad -4F^{\alpha\beta ij}F^{\gamma\delta}{}_{ij}F^{\epsilon\varepsilon kl}H_{\beta\gamma\delta}\nabla_\varepsilon F_{\alpha\epsilon kl}\,, \\[4pt]
[F^2 R\nabla H]_3 &= -2F^{\alpha\beta ij}F^{\gamma\delta}{}_{ij}R_{\gamma\delta\epsilon\varepsilon}\nabla_\beta H_\alpha{}^{\epsilon\varepsilon} + 4F^{\alpha\beta ij}F^{\gamma\delta}{}_{ij}R_{\beta\delta\epsilon\varepsilon}\nabla_\gamma H_\alpha{}^{\epsilon\varepsilon} \\
&\quad -8F_\alpha{}^\gamma{}_{ij}F^{\alpha\beta ij}R_{\gamma\delta\epsilon\varepsilon}\nabla^\varepsilon H_\beta{}^{\delta\epsilon}\,, \\[4pt]
[F^2(\nabla F)^2]_9 &= -2F_\alpha{}^\gamma{}_{ij}F^{\alpha\beta ij}\nabla_\beta F^{\delta\epsilon kl}\nabla_\gamma F_{\delta\epsilon kl} + \frac{1}{2}F^{\alpha\beta ij}F^{\gamma\delta}{}_{ij}\nabla_\epsilon F_{\gamma\delta kl}\nabla^\epsilon F_{\alpha\beta}{}^{kl} \\
&\quad +4F^{\alpha\beta ij}F^{\gamma\delta kl}\nabla_\beta F_{\delta\epsilon kl}\nabla^\epsilon F_{\alpha\gamma ij} + 4F^{\alpha\beta ij}F^{\gamma\delta kl}\nabla_\epsilon F_{\beta\delta kl}\nabla^\epsilon F_{\alpha\gamma ij} \\
&\quad -F^{\alpha\beta ij}F^{\gamma\delta}{}_{ij}\nabla_\epsilon F_{\beta\delta kl}\nabla^\epsilon F_{\alpha\gamma}{}^{kl} - 4F_\alpha{}^{\gamma kl}F^{\alpha\beta ij}\nabla_\delta F_{\gamma\epsilon kl}\nabla^\epsilon F_\beta{}^\delta{}_{ij} \\
&\quad +4F_\alpha{}^{\gamma kl}F^{\alpha\beta ij}\nabla_\epsilon F_{\gamma\delta kl}\nabla^\epsilon F_\beta{}^\delta{}_{ij} + 2F_\alpha{}^\gamma{}_{ij}F^{\alpha\beta ij}\nabla_\epsilon F_{\gamma\delta kl}\nabla^\epsilon F_\beta{}^{\delta kl} \\
&\quad +\frac{1}{4}F_{\alpha\beta ij}F^{\alpha\beta ij}\nabla_\epsilon F_{\gamma\delta kl}\nabla^\epsilon F^{\gamma\delta kl}\,, \\[4pt]
[F^2(\nabla H)^2]_5 &= -\frac{2}{3}F_\alpha{}^\gamma{}_{ij}F^{\alpha\beta ij}\nabla_\beta H^{\delta\epsilon\varepsilon}\nabla_\gamma H_{\delta\epsilon\varepsilon} + F^{\alpha\beta ij}F^{\gamma\delta}{}_{ij}\nabla_\beta H_\alpha{}^{\epsilon\varepsilon}\nabla_\delta H_{\gamma\epsilon\varepsilon} \\
&\quad +2F^{\alpha\beta ij}F^{\gamma\delta}{}_{ij}\nabla_\delta H_{\beta\epsilon\varepsilon}\nabla^\varepsilon H_{\alpha\gamma}{}^\epsilon - 2F_\alpha{}^\gamma{}_{ij}F^{\alpha\beta ij}\nabla_\epsilon H_{\gamma\delta\varepsilon}\nabla^\varepsilon H_\beta{}^{\delta\epsilon} \\
&\quad +\frac{1}{2}F_{\alpha\beta ij}F^{\alpha\beta ij}\nabla_\epsilon H_{\gamma\delta\varepsilon}\nabla^\varepsilon H^{\gamma\delta\epsilon}\,, \\[4pt]
[F^2 R^2]_4 &= -2F^{\alpha\beta ij}F^{\gamma\delta}{}_{ij}R_{\alpha\gamma}{}^{\epsilon\varepsilon}R_{\beta\delta\epsilon\varepsilon} + F^{\alpha\beta ij}F^{\gamma\delta}{}_{ij}R_{\alpha\beta}{}^{\epsilon\varepsilon}R_{\gamma\delta\epsilon\varepsilon} \\
&\quad -4F_\alpha{}^\gamma{}_{ij}F^{\alpha\beta ij}R_\beta{}^{\delta\epsilon\varepsilon}R_{\gamma\delta\epsilon\varepsilon} + \frac{1}{2}F_{\alpha\beta ij}F^{\alpha\beta ij}R_{\gamma\delta\epsilon\varepsilon}R^{\gamma\delta\epsilon\varepsilon}\,.
\end{aligned}
\tag{32}
$$

The couplings in (30) are related to the couplings in (27) or (29) by some field redefinitions, Bianchi identities, and integration by parts. However, the couplings in (30) are not manifestly invariant under T-duality, unlike the couplings in (27) or (29). The above couplings should be consistent with the S-matrix elements.

Using the tensor $t_8$ which is defined such that the contraction of $t_8$ with four arbitrary antisymmetric tensors $M^1, \cdots, M^4$ is given by [24]:

$$
\begin{aligned}
t^{\alpha\beta\gamma\delta\mu\nu\rho\sigma}M^1_{\alpha\beta}M^2_{\gamma\delta}M^3_{\mu\nu}M^4_{\rho\sigma} &= 8(\mathrm{tr}M^1M^2M^3M^4 + \mathrm{tr}M^1M^3M^2M^4 + \mathrm{tr}M^1M^3M^4M^2) \\
&\quad -2(\mathrm{tr}M^1M^2\mathrm{tr}M^3M^4 + \mathrm{tr}M^1M^3\mathrm{tr}M^2M^4 + \mathrm{tr}M^1M^4\mathrm{tr}M^2M^3),
\end{aligned}
\tag{33}
$$

one can write the couplings $[F^2 R^2]_4$ as

$$
\frac{\alpha'}{8}[F^2 R^2]_4 = -\frac{\alpha'}{32}t^{\alpha\beta\gamma\delta\mu\nu\rho\lambda}\mathrm{Tr}(F_{\alpha\beta}F_{\gamma\delta})\mathrm{Tr}(R_{\mu\nu}R_{\rho\lambda})\,.
\tag{34}
$$

Here, the trace on the Riemann curvature is over the last two indices of the Riemann curvature. The four YM couplings in the 4-derivative couplings (17) and the four Riemann couplings that the T-duality produces [25] can be written as:

$$[F^4]_4 = \frac{1}{32} t^{\alpha\beta\gamma\delta\mu\nu\rho\lambda} \text{Tr}(F_{\alpha\beta}F_{\gamma\delta})\text{Tr}(F_{\mu\nu}F_{\rho\lambda}), \tag{35}$$

$$\alpha'^2[R^4]_4 = \frac{\alpha'^2}{128} t^{\alpha\beta\gamma\delta\mu\nu\rho\lambda} \text{Tr}(R_{\alpha\beta}R_{\gamma\delta})\text{Tr}(R_{\mu\nu}R_{\rho\lambda}).$$

The above four-field couplings can be written as

$$-\frac{2\alpha'}{8\kappa^2} \int d^{10}x \sqrt{-G} e^{-2\Phi} \frac{1}{32} t^{\alpha\beta\gamma\delta\mu\nu\rho\lambda} \Big[\text{Tr}(F_{\alpha\beta}F_{\gamma\delta}) - \frac{\alpha'}{2}\text{Tr}(R_{\alpha\beta}R_{\gamma\delta})\Big] \Big[\text{Tr}(F_{\mu\nu}F_{\rho\lambda}) - \frac{\alpha'}{2}\text{Tr}(R_{\mu\nu}R_{\rho\lambda})\Big].$$

This expression has been determined in [15] through the study of the four-point S-matrix element.

# 6   Discussion

In this paper, we determine the covariant and Yang-Mills gauge invariant couplings in the classical effective action of heterotic string theory at the six-derivative order. We begin by finding the minimal basis, which consists of 435 couplings, and then impose T-duality constraints on this set. However, we find that the 435 couplings do not satisfy the T-duality constraints, indicating that the number of constraints produced by T-duality is greater than 435. We then consider the maximal basis, which contains 801 couplings, and impose the T-duality constraints on this larger set. In this case, we find that the T-duality constraints give rise to 468 relations between the coupling constants. The remaining 333 unconstrained parameters in this T-duality invariant basis can be eliminated through field redefinitions. Motivated by the observation that the T-duality constraints yield 468 relations, we also consider a basis that is neither minimal nor maximal, consisting of 468 couplings, and impose the T-duality constraints on this set. We find that all 468 T-duality constraints are satisfied by this basis, and we identify 107 non-zero couplings. Finally, we perform field redefinitions on the T-duality invariant couplings with fixed coupling constants to rewrite them in a canonical form. This results in only 85 non-zero couplings. We show that the couplings of two Riemann curvatures and two Yang-Mills field strengths are fully consistent with the results obtained from the S-matrix method [15].

To arrive at the final result in (30), we have utilized the covariance and Yang-Mills gauge invariance of the couplings in the basis. In particular, we have worked in local frames in both the external and internal spaces. In the internal space local frame, the Yang-Mills connection $A^{ij}$ is zero, while its derivatives are non-zero [4]. In this frame, the Yang-Mills field strength is given by (1), and its derivatives are ordinary covariant derivatives involving only the Levi-Civita connection. After obtaining the final result in (30), the gauge field should be replaced with the full expression:

$$F_{\mu\nu}{}^{ij} = \partial_\mu A_\nu{}^{ij} - \partial_\nu A_\mu{}^{ij} + \frac{1}{\sqrt{\alpha'}}[A_\mu{}^{ik}, A_{\nu k}{}^{j}], \tag{36}$$

and its derivatives should be replaced with derivatives that involve both the Levi-Civita and Yang-Mills connections. It is worth noting that the final result in (30) does not contain any couplings with two antisymmetric derivatives on the Yang-Mills field strength that satisfy the identity $[\nabla, \nabla]F \sim FF$. This implies that there is no ambiguity in the couplings presented in (30).

We have observed that T-duality excludes all couplings that contain the traces $\text{Tr}(FFF)$, $\text{Tr}(FFFF)$, $\text{Tr}(FFFFF)$, and their derivatives. We conjecture that this observation should be extended to the traces of all higher orders of the Yang-Mills field strength. This conjecture is consistent with the recent observation that all odd-derivative Yang-Mills gauge invariant couplings in the heterotic theory are zero [4], as such couplings involve traces of more than two $F$'s. This conjecture can be used to simplify the study of 8-derivative couplings by excluding all such couplings from the 8-derivative basis. By eliminating these terms a priori, the analysis can be streamlined and focused on the remaining, non-excluded couplings.

We have shown that the couplings in (30) with the structure $[F^2 R^2]_4$ are consistent with the 4-point sphere-level S-matrix element. Moreover, the 4-field NS-NS couplings in (31) have been demonstrated in [23] to be consistent with the corresponding S-matrix elements. These results provide confidence that the aforementioned couplings have been correctly captured. However, it is important to note that all the other couplings present in (30) should also be reproduced by the appropriate 4-point, 5-point, and 6-point functions in string theory. Performing a thorough comparison between the couplings in (30) and the corresponding higher-point string theory amplitudes would be a valuable next step. Such a detailed comparison would help to further validate the completeness and accuracy of the couplings presented in (30). It would be interesting to carry out this analysis in full detail to ensure that the final result accurately captures all the relevant contributions from string theory.

Another method for confirming the couplings in (30) is to study their cosmological reduction and validate that they satisfy the $O(d, d)$ symmetry. For the case of a vanishing Yang-Mills field, it has been shown in [26, 27, 28, 23] that the 6-derivative couplings do indeed satisfy this symmetry. It would be valuable to extend this analysis to include the Yang-Mills fields and confirm that the couplings in (30) are consistent with the $O(d, d)$ symmetry in the cosmological setting. This would provide an additional, independent check on the completeness and correctness of the couplings presented in (30).

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
