# Peer review of "Six-Derivative Yang-Mills Couplings in Heterotic String Theory"

_SciPost Physics_

## Round 1 · Referee Report · Anonymous (Referee 1) · 2024-8-15

Strengths

1) Written clearly 2) Detailed in the exposition 3) The computations are technically challenging and highly non-trivial 4) The procedure is consistent 5) The results are trustable and consistent with previous partial computations

Weaknesses

There is a technical weakness of the procedure: the output is usually given in an arbitrary scheme (set of fields) in which understanding the impact of the interactions is difficult. This is unavoidable at this stage though, but in the future one can hope to find a proper scheme in which the couplings take a simpler form.

Report

The paper contributes to a successful ongoing project by the author, intended to compute the bosonic couplings in the fist orders of the perturbative $\alpha'$ expansion of the string effective actions, by demanding the emergence of T-duality after a circle compactification. Knowing these corrections is important as they dictate the way in which string theory deforms (super)gravity.

The case in this paper corresponds to computing the six-derivative ($\alpha'^2$) interactions in the heterotic string for all the NSNS sector including the gauge fields. The strategy is essentially always the same. After the most general action is proposed, it is then simplified by using Bianchi identities, integration by parts and field redefinitions. When the action is compactified, the interactions are constrained such that the action is invariant under $\alpha'$ corrected Buscher rules up to the ambiguities mentioned previously. The method has proven to be successful so far, and allowed to compute all couplings up to $\alpha'^2$ in the bosonic string, and also gave the first proposal for the full NSNS sector completing the famous quartic Riemann interactions common to all strings. This paper extends the list by giving the result for the heterotic string to order $\alpha'^2$.

The computations are technically challenging and require the use of computer algebra, and some partial results are so lengthy that the author has omitted their inclusion in the paper. For this reason verifying or reproducing the results is difficult, but the procedure is consistent and the output has been checked to reproduce previously known results (based on supersymmetry or scattering amplitudes). I recommend the paper for publication.

Recommendation

Publish (meets expectations and criteria for this Journal)

---

## Round 1 · Referee Report · Anonymous (Referee 2) · 2024-10-3

Strengths

1 - The paper is clearly written
2 - The approach is clear and systematic
3 - The results are interesting to the community

Weaknesses

1 - The final result is quite complicated. I suspect it can be simplified considerably.

2 - In general using a computer-based approach can be very powerful, but the output is not always in a very useful form.

Report

This paper uses the requirement of T-duality invariance to determine the 6-derivative couplings in the tree-level heterotic string effective action. This is a potentially powerful approach, besides scattering amplitude calculations or supersymmetry, to fix the form of these corrections. In particular the paper determines the couplings of the gauge vectors which were not previously determined by the author.

The disadvantage of using a computer to do the calculations is that the output is not necessarily in a very useful form for humans. Indeed, I suspect that the end result can be considerably simplified. This being said I believe the paper deserves to be published in SciPost, after addressing my related comment below.

Requested changes

1 - The question of these corrections to the heterotic string was tackled by Bergshoeff and de Roo in the 80s (see in particular DOI: 10.1016/0370-2693(89)91420-2 "Supersymmetric Chern-simons Terms in Ten-dimensions"). It would be very helpful for the reader to comment on the relation to the results found there. I believe this will also suggest how the present results can be simplified.

Recommendation

Ask for minor revision

---

## Editorial Decision

resubmitted